# Diverse mechanisms of metaeffector activity in an intracellular bacterial pathogen, *Legionella pneumophila*

Malene L Urbanus[1,‡], Andrew T Quaile[2,‡] (iD), Peter J Stogios[2], Mariya Morar[2], Chitong Rao[3], Rosa Di Leo[2], Elena Evdokimova[2], Mandy Lam[4], Christina Oatway[3], Marianne E Cuff[5,6], Jerzy Osipiuk[5,6], Karolina Michalska[5,6], Boguslaw P Nocek[5,6], Mikko Taipale[3,4] (iD), Alexei Savchenko[2,6,*,†] & Alexander W Ensminger[1,3,7,**] (iD)

## Abstract

Pathogens deliver complex arsenals of translocated effector proteins to host cells during infection, but the extent to which these proteins are regulated once inside the eukaryotic cell remains poorly defined. Among all bacterial pathogens, *Legionella pneumophila* maintains the largest known set of translocated substrates, delivering over 300 proteins to the host cell via its Type IVB, Icm/Dot translocation system. Backed by a few notable examples of effector–effector regulation in *L. pneumophila*, we sought to define the extent of this phenomenon through a systematic analysis of effector–effector functional interaction. We used *Saccharomyces cerevisiae*, an established proxy for the eukaryotic host, to query > 108,000 pairwise genetic interactions between two compatible expression libraries of ~330 *L. pneumophila*-translocated substrates. While capturing all known examples of effector–effector suppression, we identify fourteen novel translocated substrates that suppress the activity of other bacterial effectors and one pair with synergistic activities. In at least nine instances, this regulation is direct—a hallmark of an emerging class of proteins called metaeffectors, or "effectors of effectors". Through detailed structural and functional analysis, we show that metaeffector activity derives from a diverse range of mechanisms, shapes evolution, and can be used to reveal important aspects of each cognate effector's function. Metaeffectors, along with other, indirect, forms of effector–effector modulation, may be a common feature of many intracellular pathogens—with unrealized potential to inform our understanding of how pathogens regulate their interactions with the host cell.

**Keywords** effector; genetic interaction; *Legionella*; metaeffector; structure-function

**Subject Categories** Chromatin, Epigenetics, Genomics & Functional Genomics; Genetics, Gene Therapy & Genetic Disease; Microbiology, Virology & Host Pathogen Interaction

**Mol Syst Biol. (2016) 12: 893**

## Introduction

The concept of effector-based modulation of host pathways is central to the current molecular understanding of microbial pathogenesis. In this view, effector activity is directed against host factors and is regulated by changes to effector expression or through modulation of translocation efficiency. There are, however, several ways that translocated proteins might functionally interact once inside the host (Shames & Finlay, 2012). The regulatory complexity provided by effector–effector interactions may add another layer to the program of pathogenesis, helping balance host perturbation with the maintenance of a replicative niche. Pathogens that strongly rely on their hosts for replication and translocate large numbers of proteins are candidates for possessing such effector complexity (Ensminger, 2016). The intracellular bacterial pathogen *L. pneumophila* uses over 300 Icm/Dot-translocated substrate [IDTS; Table EV1, for review, see Ensminger (2016); Franco *et al* (2009); Isaac and Isberg (2014)] to grow within protozoan and mammalian hosts (Rowbotham, 1980; Fields, 1996; Molmeret *et al*, 2005). A few

1   Department of Biochemistry, University of Toronto, Toronto, ON, Canada
2   Department of Chemical Engineering and Applied Chemistry, University of Toronto, Toronto, ON, Canada
3   Department of Molecular Genetics, University of Toronto, Toronto, ON, Canada
4   Donnelly Centre for Cellular and Biomolecular Research, University of Toronto, Toronto, ON, Canada
5   Biosciences Division, Argonne National Laboratory, Structural Biology Center, Lemont, IL, USA
6   Midwest Center for Structural Genomics, Lemont, IL, USA
7   Public Health Ontario, Toronto, ON, Canada
    *Corresponding author. Tel: +1 403 210 7980; E-mail: alexei.savchenko@utoronto.ca
    **Corresponding author. Tel: +1 416 978 6522; E-mail: alex.ensminger@utoronto.ca
    ‡These authors contributed equally to this work
    †Present address: Department of Microbiology, Immunology and Infectious Diseases, Cumming School of Medicine, University of Calgary, Calgary, AB, Canada

notable examples of effector–effector functional interaction have been documented in *L. pneumophila*, in which effectors directly or indirectly regulate the activity of each other inside the eukaryotic cell (Kubori *et al*, 2010; Neunuebel *et al*, 2011; Tan & Luo, 2011; Tan *et al*, 2011; Havey & Roy, 2015; Jeong *et al*, 2015).

Effector–effector interactions have largely been identified through individual studies of effector function; nevertheless, several early examples provide useful insight into what we might expect to find within a more complete set of such interactions. These interactions could take the form of direct effector–effector suppression, such as the case with the *L. pneumophila* IDTS LubX. LubX is an E3 ubiquitin ligase that targets another translocated substrate (SidH) for proteasomal degradation during the late stages of intracellular replication and is the founding member of a group of proteins known as "metaeffectors"—so named because it is an "effector of effectors" (Kubori *et al*, 2010). In addition to this rare example of direct effector–effector interaction, other translocated substrates are known to indirectly antagonize one another by targeting the same host proteins or pathways with counteracting activities. For instance, the *L. pneumophila* IDTS AnkX adds a phosphocholine group to a host protein, Rab1 (Mukherjee *et al*, 2011), while another IDTS, Lem3, removes it as part of a regulatory cascade (Tan *et al*, 2011). Similarly, SidM adds an AMP moiety to Rab1 (Muller *et al*, 2010) that is removed by the antagonizing IDTS SidD (Neunuebel *et al*, 2011; Tan & Luo, 2011). SidJ was also recently shown to be a functional antagonist for the SidE family of effectors (Havey & Roy, 2015; Jeong *et al*, 2015), which have a unique non-canonical ubiquitination activity through their mono-ADP-ribosyltransferase motif that does not require components of the E1, E2, and E3 ubiquitin enzyme cascade (Qiu *et al*, 2016). SidJ releases SidE and its paralogs from the *Legionella*-containing vacuole membrane, though the exact mechanism of the SidJ-dependent release remains unknown (Jeong *et al*, 2015). As this phenomenon remained to be explored systematically, we screened over 108,000 pairwise effector–effector genetic interactions between two libraries of ~330 effectors co-expressed in *Saccharomyces cerevisiae*—an established high-throughput proxy for the eukaryotic cell.

# Results

### Widespread functional antagonism between IDTS

To systematically map IDTS functional interactions, we sought a high-throughput, genetically tractable proxy for the diverse eukaryotic hosts within which the bacteria normally replicate. Because IDTS largely targets highly conserved eukaryotic pathways (a consequence of their dependence on diverse protozoa in the environment), the expression of several individual IDTS within the yeast *S. cerevisiae* is known to cause cell growth defects (Campodonico *et al*, 2005; Shohdy *et al*, 2005; de Felipe *et al*, 2008; Heidtman *et al*, 2009; Xu *et al*, 2010; Belyi *et al*, 2012; Guo *et al*, 2014; Nevo *et al*, 2014). Over 200 IDTS remained uncharacterized in yeast, so we generated a library of 330 yeast strains in which each strain expresses one IDTS (or putative IDTS) on a high-copy HIS3 plasmid under control of a galactose-inducible promoter. We measured their growth by high-density spot size and liquid growth curves (Appendix Fig S1A, Table EV1). In these assays, 227 proteins

conferred a moderate to severe growth defect in yeast when expressed, presenting a clear opportunity to identify effector–effector interaction by screening for suppression of these growth defects.

In order to obtain an IDTS-wide genetic interaction map, we modified the automated, high-density replica plating approaches previously developed for analyzing *S. cerevisiae* double mutants through synthetic genetic array (SGA) analysis (Tong *et al*, 2001). Instead of looking at double mutants, however, we used yeast genetics to systematically assess the effects of effector co-expression on yeast growth (Fig 1A, see also Materials and Methods). Critically, a previously characterized yeast library (Heidtman *et al*, 2009 and Appendix Fig S1B) had properties (galactose-inducible, URA+, MATα) that made it inherently compatible with the HIS+, MATa library described above (Appendix Fig S1A, Table EV1). After extending the Heidtman collection by over 200 clones (Table EV7), we used a pinning robot to mate each strain from this collection to an ordered array of our HIS3 plasmid effector library (laid out in quadruplicate in a 1,536-spot pinning density). We selected for diploids on media lacking uracil and histidine and then replica plated them onto selective medium containing galactose to induce expression from both plasmids. After 2 days of incubation, we used automated image analysis to quantify the spot size of strains expressing each of the > 108,000 possible pairwise combinations between these libraries (Fig 1A).

Using this approach, we identified all previously described instances of effector–effector antagonism (Fig 1B, shaded), as well as several IDTS whose growth inhibition in yeast was suppressed by one or more previously unknown antagonists. Due to technical reasons, growth suppression was not always observed in both directions (when the identity of an array and query strain were reversed). While infrequent, such instances are likely due to the potential of epitope tags and spontaneous mutations within the yeast genome to mask some interactions. As such, we also looked for additional suppressors in which expression of a query gene was able to suppress growth inhibition caused by one of the IDTS on the array (Fig 1C, Appendix Fig S2). In total, eighteen IDTS suppress the deleterious effects of one or more of the other effectors within the eukaryotic cell. Notably, all of the effector–effector interactions that we observe occur between experimentally validated translocated substrates (Table EV1). While each of the previously characterized effector–effector functional interactions involved effectors chromosomally adjacent to one another (or the broad-acting paralogs of SidJ and SidE), several of our newly identified pairs are genomically unlinked.

### Identification of inter-substrate physical interaction

Effector–effector suppression could reflect counteracting activities on a shared host target (or pathway) or it could reflect the direct modulation of one IDTS by the activity of another. To identify instances of direct suppression, we examined all 23 effector–effector suppression pairs for physical interactions using the yeast two-hybrid (Y2H) assay (Fields & Song, 1989; Fig 2A) and used an established high-throughput mammalian system, LUMIER (Barrios-Rodiles *et al*, 2005; Taipale *et al*, 2012) to measure all possible physical interactions between the cohort of functional antagonists and their cognate IDTS (Fig 2B). A previous study comparing Y2H,

**Figure 1.**

LUMIER, and other assays showed that each technology captures only a subset of true complexes (Braun *et al*, 2009), justifying the application of multiple approaches. In total, we identified nine novel effector–effector complexes (Table EV2), not including the interaction between LubX and SidH that was previously identified using purified proteins (Kubori *et al*, 2010) and by affinity purification (Quaile *et al*, 2015). This is not surprising, as LubX targets SidH for degradation—and raises the likely possibility that even more effector–effector complexes may be identified through additional methods.

Though SidJ was classified as a metaeffector of SidE and its paralogs due to its impact on their localization during infection (Jeong *et al*, 2015), previous studies were unable to show a direct interaction between SidJ and SdeA (Havey & Roy, 2015; Jeong *et al*, 2015). Notably, we detect a SdeC and SidJ complex in mammalian cells, suggesting that it is possible that SidJ regulates at least one SidE family representative through a direct physical interaction. Interestingly, while SidJ suppresses the activity of SidE, SdeB, and SdeC in our assays, its paralog SdjA (Liu & Luo, 2007) is a much more potent regulator of these SidE-family members. We detect complexes of SdjA with SdeB and SdeC by the LUMIER assay, suggesting further attention should be placed upon the functional overlap and potential diversification of each of the SidJ and SidE paralogs.

### A multifunctional metaeffector SidP links its cognate effector, MavQ, to PIP modulation

We were surprised that one of the putative metaeffectors identified in our screen, SidP, has an established role as a phosphatidylinositol 3-phosphate (PI3P) phosphatase (Toulabi *et al*, 2013). In addition to this canonical role against the host, SidP also inactivates an effector of unknown function, MavQ (Fig 1B) and does so via a physical interaction—as shown by both Y2H (Fig 2A) and AP-MS (Table EV3).

SidP has three distinct domains: an N-terminal phosphatase domain along with insertion (I) and C-terminal domains (Toulabi *et al*, 2013) each of thus far unknown functions (Fig 3A). Catalytically inactive SidP mutants retained the ability to suppress MavQ, indicating that the mechanism of SidP metaeffector activity is independent of its role as a PI3P phosphatase against the host. In contrast, the previously uncharacterized C-terminal domain of SidP was both necessary and sufficient for MavQ inactivation and binding (Fig 3B and Appendix Fig S3B).

Using a guilt-by-association approach, we asked whether MavQ also plays a role in the regulation of PI metabolism during infection. If so, SidP's metaeffector activity against MavQ could serve to coordinate the modulation of lipid metabolism during infection. HHpred analysis (Soding, 2005) of MavQ reveals a weak similarity to PIP kinases, suggesting several of its N-terminal residues might be involved in ATP binding and catalysis (Fig 3C). To test this, we individually mutated MavQ residues corresponding to the active site residues in PIP kinases. D147A and D160A substitutions completely abrogate yeast growth inhibition (Fig 3D), consistent with their predicted role in the active site. Using an *in vitro* activity assay, we observe that wild-type MavQ, but not the D147A mutant, possesses robust ATP hydrolysis activity (Fig 3E). Importantly, the addition of phosphoinositide to this reaction increased the activity of the wild-type MavQ fivefold as would be expected of a PIP kinase. Furthermore, like SidP, we also observe robust MavQ binding to a broad range of PIP molecules using the protein–lipid overlay assay (Appendix Fig S3D). Taken together, these data support a model in which MavQ is a kinase with a role in PIP modulation. In this context, the inactivation of MavQ by the C-terminal domain of SidP likely provides a molecular mechanism by which to directly coordinate the activities of these two proteins against host PIPs.

### The metaeffector LegL1 functions by blocking the active site of its cognate effector, RavJ

Our discovery of several novel direct IDTS antagonists in the *L. pneumophila* genome also provided an opportunity to start to define the full scope of mechanisms that a bacterial pathogen can use to regulate its own effectors after release into the host cell. To that end, we performed detailed structure-function analysis of a completely uncharacterized set of proteins: the putative metaeffector LegL1, a leucine-rich repeat (LRR) protein (de Felipe *et al*, 2005), and its cognate effector, RavJ.

To define the mechanism of LegL1 inhibition of RavJ, we first solved the crystal structure of the N- and C-terminal portions of RavJ spanning amino acid residues 1–228 and 251–371, respectively (Fig 4A, Table EV4). Despite no identifiable sequence similarity, the overall fold of the N-terminal domain of RavJ resembled that of papain-like cysteine protease family members (Fig 4A). Comparison of RavJ with its top structural homolog, human tissue

**Figure 1. Effector–effector functional interactions.**

A  High-throughput effector–effector suppression profiling in yeast. Strains carrying an individual IDTS expressed from an inducible plasmid are mated into a 384-array of strains carrying a compatible library of 330 IDTSs. The diploid array is pinned onto galactose medium to induce expression and spot size measured through image analysis. A 1,536-spot array (SidM query) after 2 days of growth is shown.

B  Suppression of growth defects through inter-effector antagonism. Screening the > 108,000 pairwise IDTS interactions possible between the libraries reveals sixteen suppression profiles. Spot sizes of strains co-expressing the IDTS query clone (in red) and each of the 330 effectors within the compatible arrayed IDTS library were quantified for two independent biological replicate screens and plotted on separate axes. Error bars represent standard deviation (SD) of spot size between quadruplicate spots measured within each biological replicate (see Materials and Methods). *Bona fide* suppressors show consistent divergence from the population following a diagonal line, while spurious yeast suppressor mutations that sometimes appear for toxic queries (e.g., SidL/Ceg14 and SdbB) occur in one replicate with a large SD. The shaded box contains all previously known suppression pairs, plus the novel antagonist, SdjA (*) of SidE, SdeB, and SdeC. "lc" = screened using a low-copy query due to an inability to clone the IDTS in high-copy vectors.

C  Suppression of array strain growth defects by specific queries reveals additional instances of effector inactivation. Array strains were mated to create either query-IDTS-containing diploids (*y*-axis) or empty vector-IDTS diploids (*x*-axis). In this representation, suppression of a growth inhibitory array strain is revealed as a deviation above the diagonal. Error bars represent the SD of spot size between quadruplicate spots within each dataset. (Note that the larger spread of diploid spot sizes in panel C reflects that, unlike in panel B, these queries are not toxic on their own.)

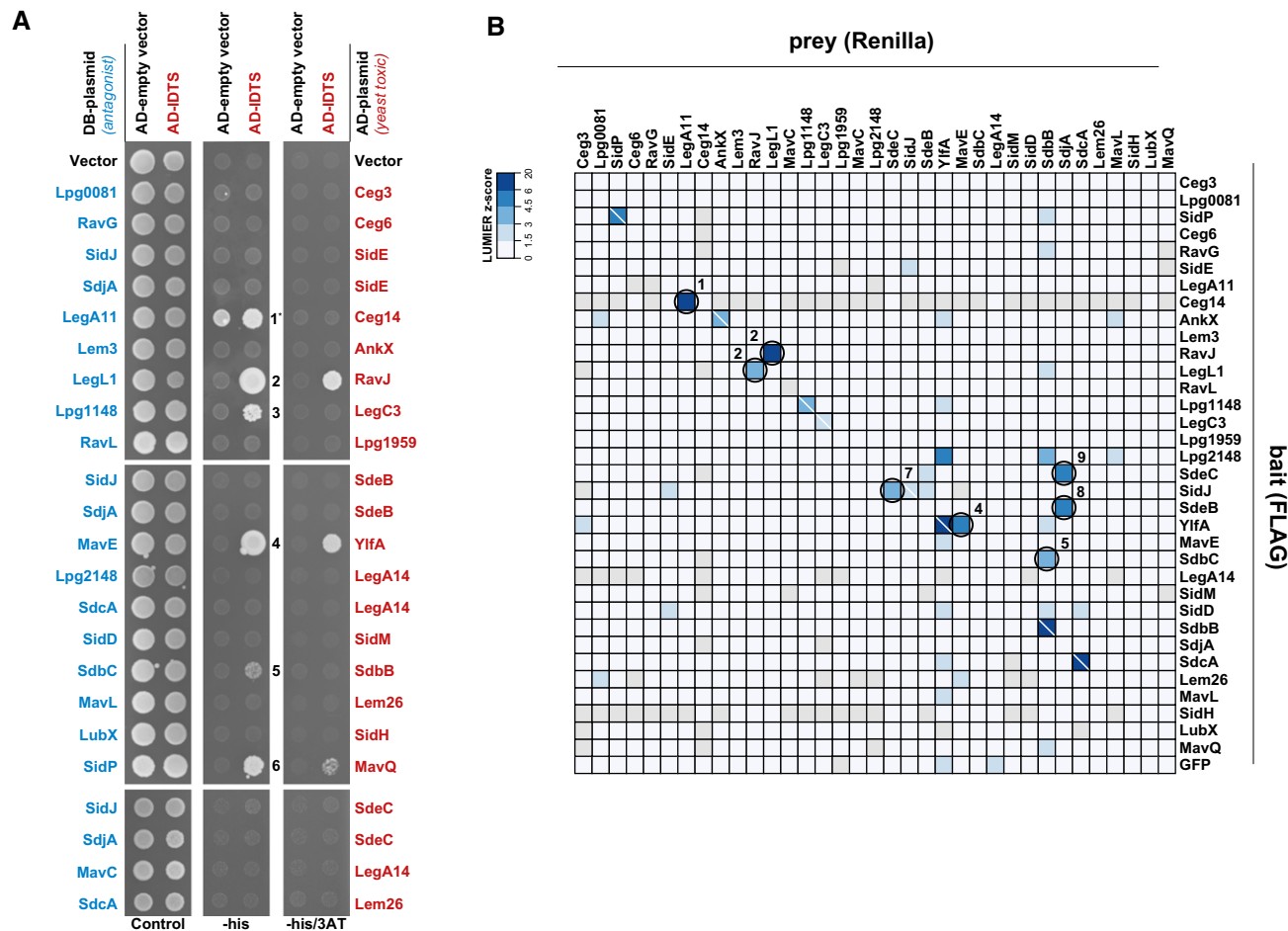

**Figure 2. Effector–effector physical interactions in yeast and mammalian cells.**

A Direct physical interactions between effectors as revealed by yeast two-hybrid (Y2H) assay. For each of the 23 effector–effector suppression pairs, the rescuer was fused to the Gal4 DNA-binding domain (DB) and its cognate toxic IDTS was fused to the Gal4-activating domain (AD). DB-IDTS with an AD-empty vector control or AD-IDTS were tested for growth on control medium or medium lacking histidine −/+ 3AT. Physical interactions drive *HIS3* expression and confer growth under −his (low stringency) or −his +3AT (high stringency) conditions. Autoactivation of DB-IDTS (the ability to grow on selective conditions without a DB-AD complex) was observed for only one IDTS, LegA11.

B The LUMIER assay was used to detect physical interactions in mammalian cells. FLAG-V5-tagged bait proteins (*y*-axis) and *Renilla* luciferase-tagged prey proteins (*x*-axis) were co-transfected into HEK293T cells. Lysates were added to anti-FLAG-treated 384-well plates. After washing, co-precipitation is detected as a luminescent signal. Shown are luminescence Z-scores calculated over all *Renilla* luciferase-tagged preys. FLAG-V5-bait proteins that were not expressed were filtered from the dataset and are shown in gray. A white backslash indicates a homo-dimer interaction. Circles indicate functionally antagonistic pairs with a Z-score > 3. Numbers indicate the 9 novel effector–effector complexes that were identified using the Y2H and LUMIER methods (Table EV2).

transglutaminase (TGM2), identified three conserved residues (C101, H138, D170) along with another adjacent residue (W172) as a putative catalytic site (Appendix Fig S4A and B). The C-terminal domain of RavJ did not reveal any structural similarity to known proteins or functional domains, but we identified a short conserved motif (H316-WNRHH-V322; Fig 4A) that is present in another IDTS, WipB, as well as several non-*Legionella* proteins. To determine which of these structural elements are essential for effector function, we tested the effect of their ablation on yeast growth. Strikingly, single substitutions in each of the four N-terminal catalytic site residues, as well as the C-terminal loop mutant, abrogated the yeast growth inhibition of RavJ (Fig 4B).

After using Y2H and gel filtration analysis to demonstrate a physical interaction between LegL1 and the N-terminal domain

of RavJ (Appendix Fig S5A–C), we cocrystallized and determined the crystal structure of this novel effector–effector complex (Fig 4C, Table EV4). The heterocomplex has a 1:1 stoichiometry, with LegL1 forming a canonical LRR horseshoe-shaped structure arching over and interacting with the RavJ active site identified above. Such intimate contact suggests that LegL1 is able to block the activity of RavJ by sterically hindering its catalytic site. This structure, to our knowledge, represents the first metaeffector–effector complex structure, revealing a new archetype of metaeffector activity: non-proteolytic inter-substrate inhibition.

To confirm the modularity of RavJ suggested above, we used AP-MS analysis to identify its host targets. The C-terminal domain of RavJ was both necessary and sufficient to interact

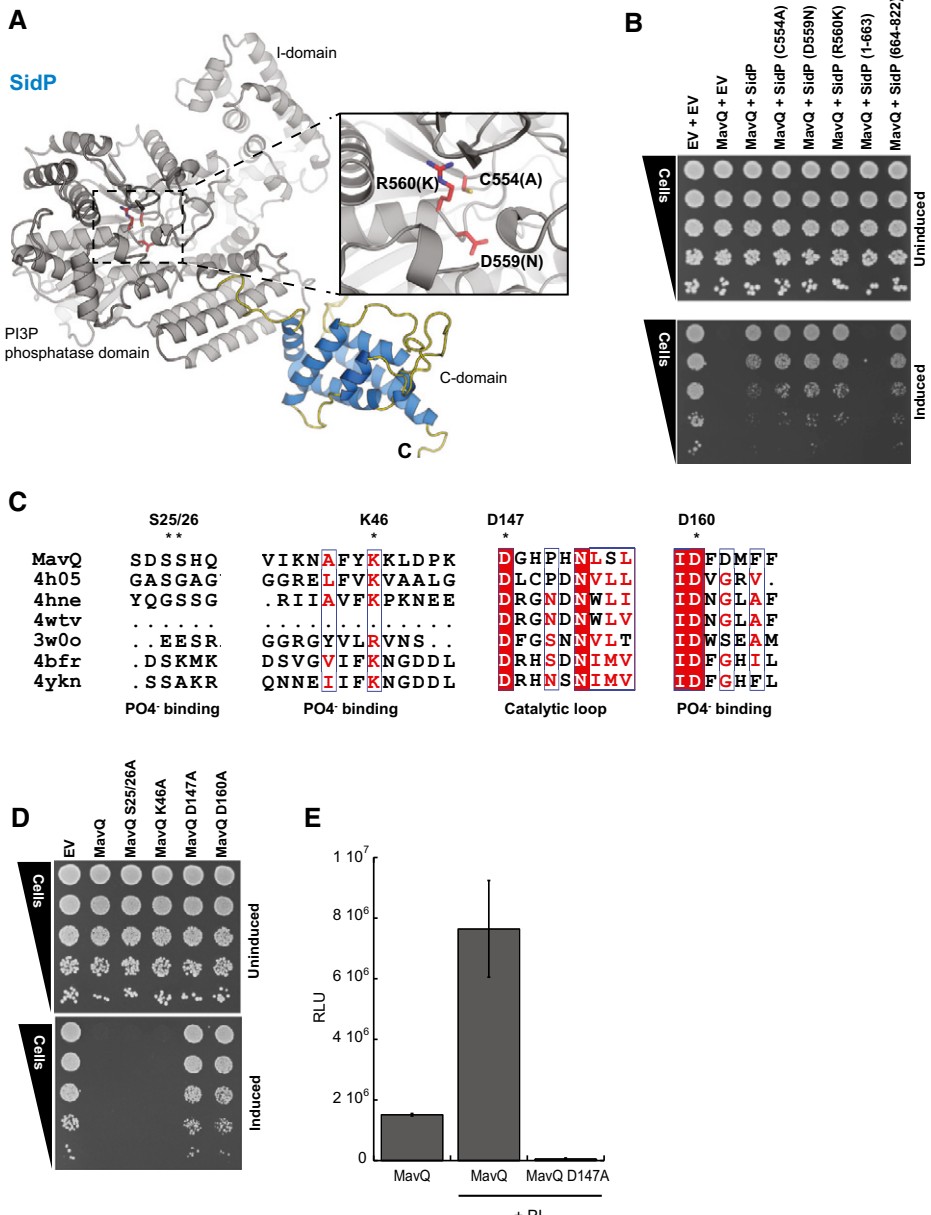

**Figure 3.  A link between the uncharacterized IDTS MavQ, SidP, and PIP modulation.**

A    SidP (Lpg0130) modeled by Phyre2 on the published structure of its *Legionella longbeachae* ortholog (4JZA; Toulabi *et al*, 2013). Phosphatase catalytic site residues and mutants previously shown to abolish activity (Toulabi *et al*, 2013) are highlighted.

B    The C-terminal domain of SidP is required and sufficient for MavQ inactivation. SidP phosphatase mutants (C554A, D559N, and R560K, see inset A) and SidP fragments (1–663 and 664–822) were tested for the ability to inactivate MavQ in a yeast spot dilution assay on glucose (uninduced, upper panel) or galactose (induced, lower panel).

C    HHpred (Soding, 2005) suggests that the N-terminal part of MavQ may share homology with several PI3 and PI4 kinases and aminoglycoside phosphotransferases (see Materials and Methods for more information). The phosphate binding and catalytic loop regions are shown with identical residues (red) and similar residues (white box) highlighted. The putative ATP binding site and catalytic aspartate residues are indicated (*).

D    MavQ residues S25, S26, K46, D147, and D160, which correspond to PI4P kinase residues involved in ATP binding and catalysis, were mutated and tested in a yeast spot dilution. The D147A and D160A mutants abrogate yeast growth inhibition by MavQ. Each mutant was tested for expression and stability (Appendix Fig S3C).

E    *In vitro* ATP-to-ADP conversion by MavQ or MavQ D147A was measured in the presence of PI micelles using the ADP-Glo kinase assay (Zhou *et al*, 2014). Basal ATP-to-ADP conversion increases several fold in the presence of PI micelles. Mutation of D147 in MavQ ablates this activity as the mutant activity is significantly different from wild type as assessed by an unpaired, two-tailed Student's *t*-test (P-value = 0.02, n = 2). Error bars indicate the SD.

with several components of the eukaryotic septin and elongator complexes (Fig 4D). Mutation of the C-terminal conserved motif (H316A|R319A|H320A, "loop mt") was sufficient to ablate the interaction with both complexes—in agreement with our observation that this mutant and deletion of the entire C-terminal domain cause a similar alleviation of the yeast growth defect.

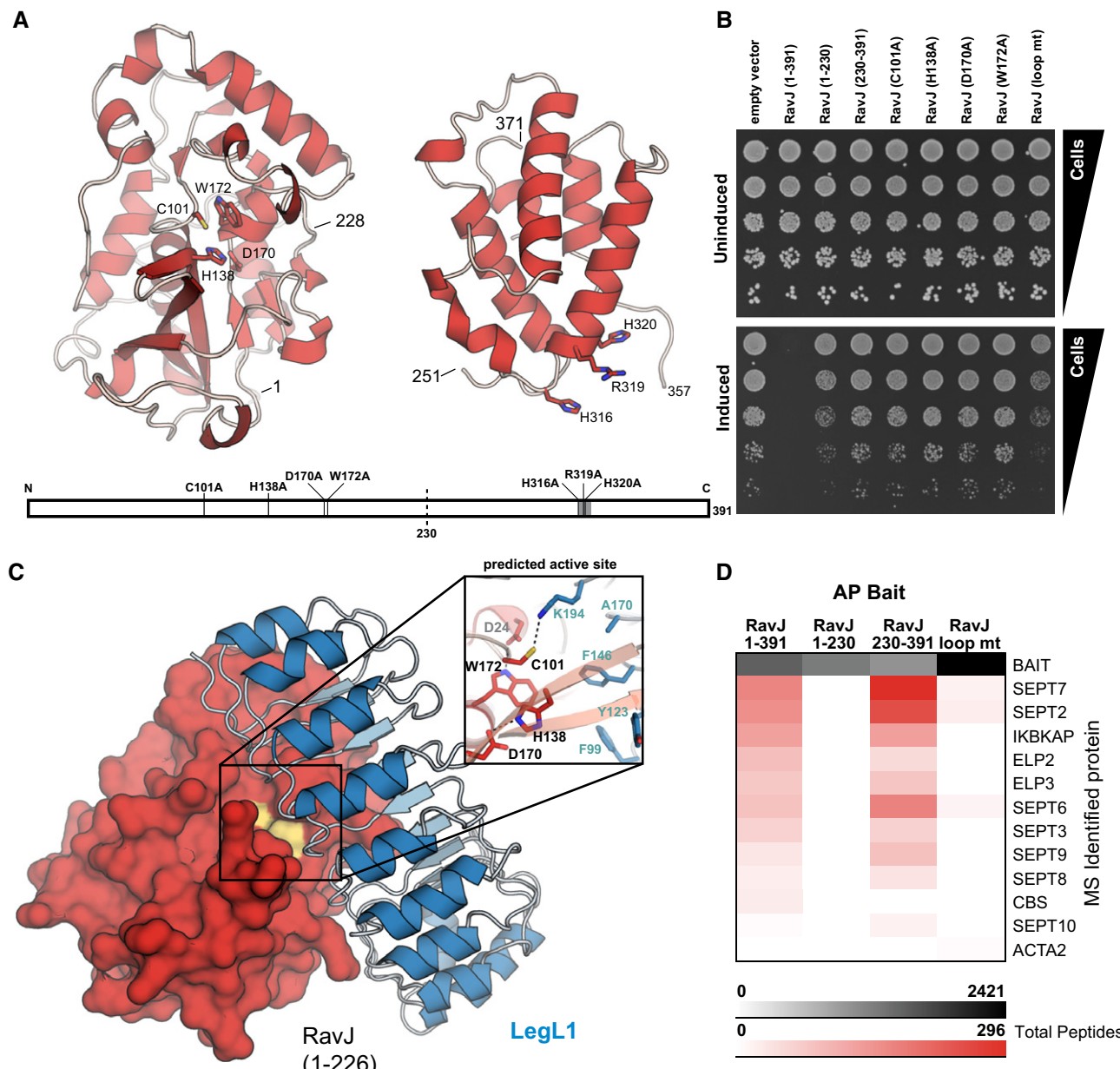

**Figure 4.  RavJ modularity and inhibition by LegL1.**

A   Structures for amino acid residues 1–228 and 251–371 of RavJ were solved separately to a resolution of 1.3 and 1.9 Å, respectively. The N-terminal structure contains a structurally conserved putative catalytic triad (C101/H138/D170). A conserved motif (highlighted in gray) is located in the C-terminal domain with surface exposed residues H316, R319, and H320.

B   While expression of full-length RavJ showed a severe growth defect in a spot dilution assay, mutation of each of the putative catalytic residues or the adjacent W172 to alanine fully relieved toxicity as did substitutions in the conserved C-terminal motif. Each mutant was tested for expression and stability (Appendix Fig S4C).

C   LegL1 acts as a direct antagonist by blocking the active site of RavJ. The co-crystal complex of LegL1 (blue) bound to RavJ (1–226, red) was solved to a resolution of 2.0 Å. LegL1 forms a canonical leucine-rich repeat structure arching over the predicted active site of RavJ. The interface spans 1,240 Å$^2$ and is reinforced by LegL1 K194 projecting into the RavJ active site where it forms a hydrogen bond with the predicted catalytic residue C101 (inset).

D   Immobilized RavJ was incubated with U937 cell lysate and interacting proteins were identified by nLC-MS/MS. Each column in the table represents the sum of the average total peptide counts for two replicates of affinity purification-mass spectrometry. Septins and elongator complex proteins were identified only with full-length RavJ and the (230–391) C-terminal domain, suggesting that the C-terminal domain is a substrate-binding domain. Use of the otherwise full-length loop mutant (H316A/R319A/H320A) as bait abrogated these interactions.

The RavJ-septin AP-MS data were confirmed by co-IP (Appendix Fig S5D). Taken together, our data reveal RavJ as a two-domain effector, in which the C-terminal domain is responsible for host protein recognition, while the N-terminal domain carries catalytic activity specifically inhibited by direct binding of the metaeffector LegL1.

### The deubiquitinase LupA inactivates its cognate effector LegC3 through removal of ubiquitin

We next solved the crystal structure of another novel metaeffector, the previously uncharacterized Lpg1148, to reveal a domain (residues 123–304, Fig 5A, Table EV4) typical of eukaryotic ubiquitin proteases (UBP) involved in the deconjugation of ubiquitin or ubiquitin-like proteins from their targets (Appendix Figs S6A and B). As Lpg1148 rescues the yeast growth defect caused by LegC3 (Fig 1C), we hypothesized that this inhibition may derive from the catalytic functionality revealed by our structure.

*In vitro* activity assays confirmed that Lpg1148 is a ubiquitin-specific protease (Fig 5B) whose activity is abrogated by a mutation within the predicted catalytic triad (Fig 5C). We therefore hypothesized that Lpg1148 (hereafter referred to as *Legionella* ubiquitin-specific protease A—LupA) removes a ubiquitin modification from LegC3 that otherwise supports its activity in a proteasomal-independent manner. As a test of this hypothesis, we co-transfected human HEK293T cells with LegC3 and either wild-type or

catalytically impaired variants of LupA. In accordance with our model for LupA metaeffector activity, we detected ubiquitinated species of LegC3 in the presence of catalytically inactive LupA variants but not in the presence of the wild-type deubiquitinase (Fig 5D). These data reveal another novel mechanism of direct effector inactivation—the specific deubiquitination of a cognate effector—and suggest that LegC3 activity may depend upon modification by an endogenous (human) E3 ligase.

### Evolutionary implications of direct effector–effector suppression

We next tested whether direct effector antagonists might be distinguished within pathogen arsenals based on the capacity for co-evolution of the effector–effector interface. We reasoned that while direct antagonists and their cognate effectors are likely to co-evolve during *Legionella* speciation (Fig 6A), effectors that antagonize one another indirectly through a common host target are under evolutionary pressure to maintain their individual activities against the conserved protozoan orthologs of that target (Fig 6B). One

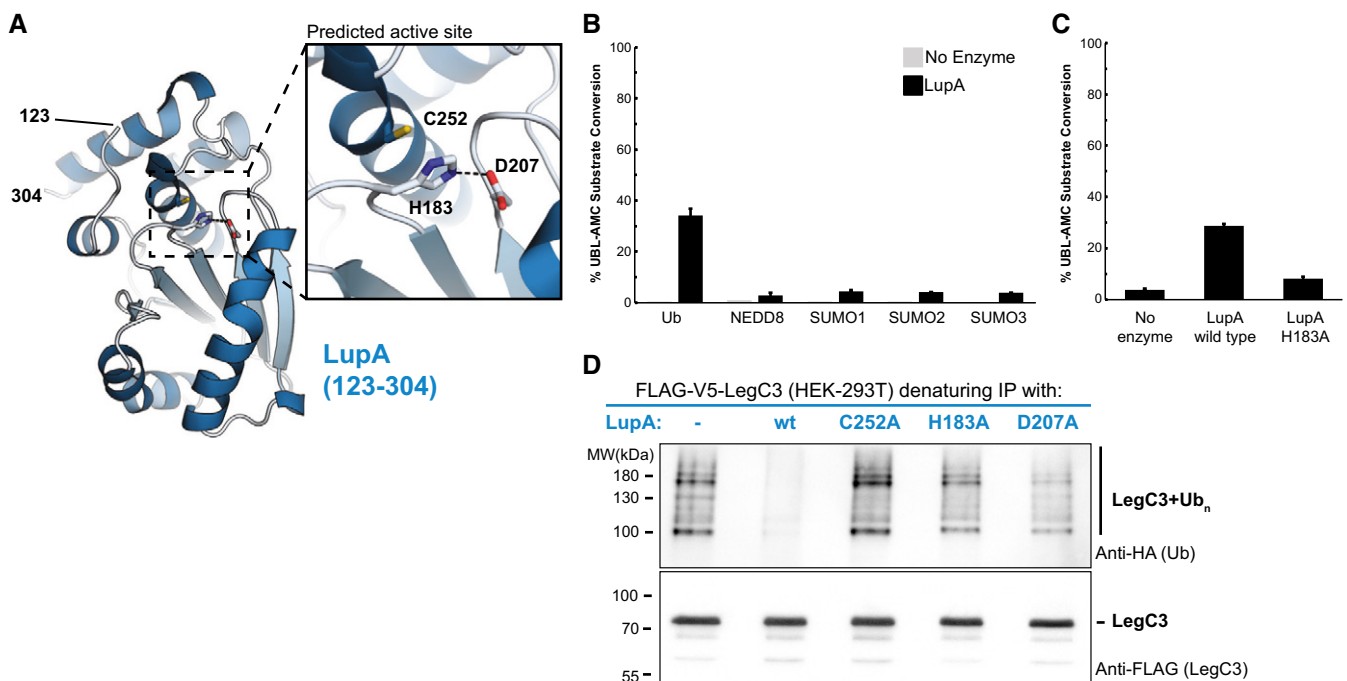

**Figure 5. Metaeffector activity through deubiquitination.**

A   The *de novo* crystal structure of the metaeffector LupA (123–304) was determined to 1.9 Å resolution and reveals that LupA belongs to the ubiquitin or ubiquitin-like protease (UBP) family of proteins with a canonical cysteine protease triad (inset).

B   LupA displays deubiquitinase activity *in vitro*. A fluorescence-based assay was used to monitor the catalytic hydrolysis of ubiquitin and ubiquitin-like proteins from a covalently linked fluorophore (AMC) after incubation with purified LupA or a no-enzyme control. Substrates tested are displayed on the *x*-axis, and % substrate conversion on the *y*-axis. LupA activity is specific toward ubiquitin. The activity of LupA against ubiquitin-AMC is significantly different than its activity against each of the other ubiquitin-like substrates as assessed by unpaired, two-tailed Student's *t*-tests (Ub versus Nedd8: *P*-value = 0.005; SUMO-1: *P*-value = 0.005; SUMO-2: *P*-value = 0.005; SUMO-3: *P*-value = 0.005; *n* = 2). The error bars indicate the SD.

C   Mutation of a predicted catalytic residue (H183) almost completely abolishes the *in vitro* hydrolase activity. In the fluorescence-based assay described above, the mutant activity is significantly reduced from the wild-type enzyme activity as assessed by an unpaired, two-tailed Student's *t*-test (*P*-value = 0.000009, *n* = 3). The error bars indicate the SD.

D   LupA activity removes polyubiquitin linkages from its cognate IDTS, LegC3. Denaturing IPs of FLAG-V5-LegC3 expressed in HEK293T cells were analyzed by western blot and probed for ubiquitination during co-expression of V5-LupA or one of three catalytically impaired variants (see Appendix Fig S6C for input). While polyubiquitination of LegC3 was present when co-expressed with a catalytically inactive LupA (or in the absence of LupA), no ubiquitination could be detected in the presence of wild-type LupA, confirming its deubiquitination of LegC3 *in vivo*.

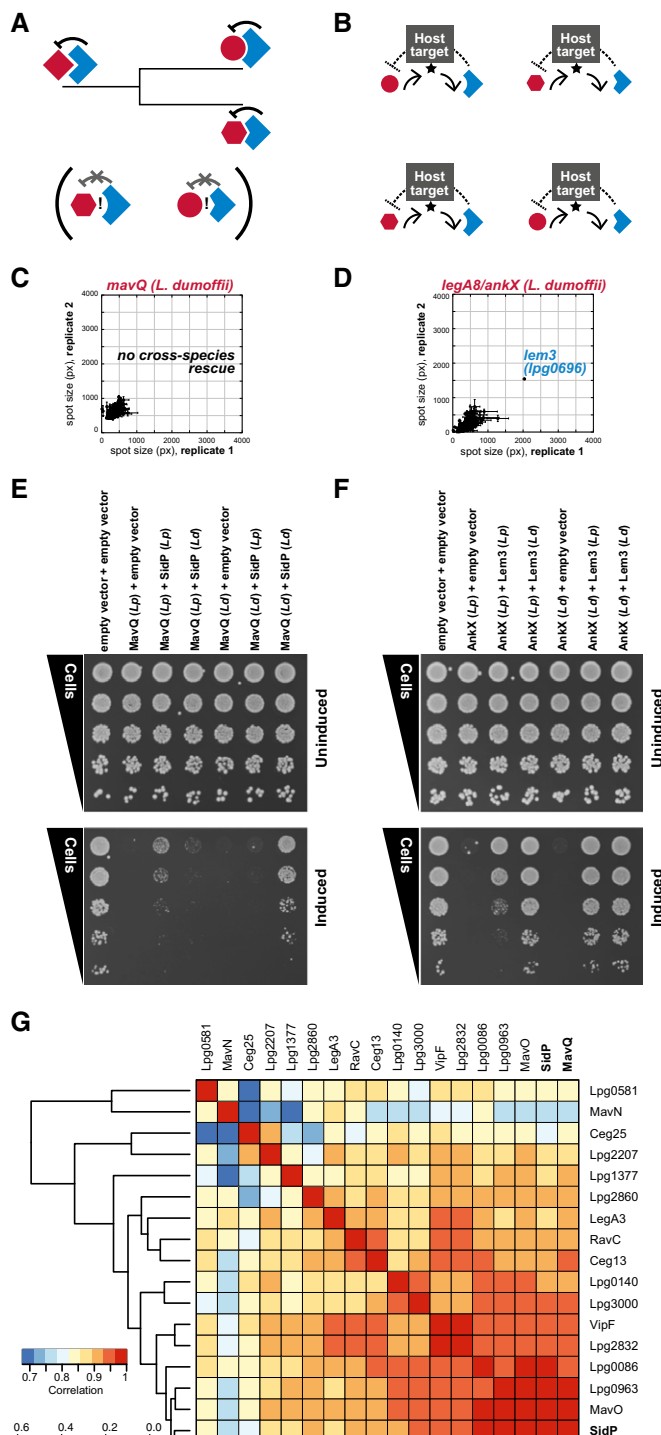

**Figure 6.  The evolutionary constraints of inter-substrate antagonism.**

A   Functional antagonism by direct physical interactions between an antagonist (blue) and its bacterial target (red) provides an opportunity for co-evolution of each direct antagonist and its cognate translocated substrate during bacterial speciation. A prediction of this model is that (i) direct antagonist activity is likely to be maintained within species, but (ii) co-evolution is likely to modify the protein–protein interface such that cross-species rescue between a direct antagonist and the ortholog of its cognate substrate is unlikely.

B   In contrast, indirect functional antagonism that occurs through counteracting activities on a shared host target is likely to be maintained cross-species.

C   Consistent with this model, direct antagonist activity is not maintained between orthologs derived from distinct *Legionella* species as revealed by screening the *L. pneumophila* library using an ortholog of SidP from *L. dumoffii*. Error bars represent the SD of spot size between quadruplicate spots measured within each biological replicate.

D   Indirect antagonism is maintained, as revealed by the rescue of an *L. dumoffii* AnkX query with *L. pneumophila* Lem3 on the array. Error bars represent the SD of spot size between quadruplicate spots measured within each biological replicate.

E, F   Intra-species rescue is maintained in *L. dumoffii* for both types of antagonists. The antagonist pairs MavQ-SidP (E) and AnkX-Lem3 (F) from *L pneumophila* (*Lp*) and *L. dumoffii* (*Ld*) were tested for intra- and inter-species rescue by co-expressing orthologs from the same species (*Ld+Ld* or *Lp+Lp*) and orthologs from different species (*Lp+Ld* or *Ld+Lp*) using a spot dilution assay on glucose (uninduced, upper panel) and galactose (induced, lower panel). Inter-species rescue was only observed for AnkX and Lem3, consistent with the results in (A). Both pairs showed intra-species rescue.

G   Genomic analysis uncovers signatures of MavQ-SidP co-evolution. We examined the phylogenies of MavQ and SidP across 35 species, along with all the other proteins in our library with one and only one ortholog in the same set. (For technical issues shaping species choice, please see Materials and Methods). MirrorTree analysis was performed to generate inter-protein correlation scores based on the phylogenetic trees of each of these IDTSs. Hierarchical clustering analysis of these scores places SidP and MavQ in a linked group of highly correlated phylogenies.

prediction of this model is that indirect functional antagonists from distinct *Legionella* species will maintain their ability to counteract one another's activity, whereas a direct antagonist from one species is likely to only suppress the activity of its cognate effector within that species.

To test this hypothesis, we cloned orthologs of: (i) indirect antagonists AnkX and Lem3 and (ii) direct antagonists MavQ and SidP from *Legionella dumoffii*, a related species of *Legionella* that is also

able to cause human disease (Qin *et al*, 2012). We observe no cross-species rescue on the array between MavQ (*L. dumoffii*) and our *L. pneumophila* library (Fig 6C), consistent with a divergence of the MavQ-SidP interface since these two *Legionella* species' last common ancestor. In contrast, AnkX cloned from *L. dumoffii* was efficiently rescued by the *L. pneumophila* ortholog of Lem3 on the array (Fig 6D). As predicted, intra-species rescue was maintained for both gene pairs (Fig 6E and F). Notably, Y2H assays confirmed a direct physical interaction between the *L. dumoffii* orthologs of MavQ and SidP (Appendix Fig S7), consistent with a conserved role of SidP as a direct antagonist of MavQ. (A technical aside: the standard Y2H assay relies on expression of a growth-based reporter (*HIS3*) and resultant growth on histidine-deficient media; thus, any examination of cross-species physical interaction using this approach would be obscured by MavQ's unchecked growth inhibition.)

To look for further evidence that MavQ and SidP may have co-evolved during *Legionella* speciation, we performed MirrorTree analysis (Pazos & Valencia, 2001) to compare the phylogenetic trees of MavQ, SidP, and the other effectors in our library for which pairwise orthologs could be identified across a shared set of 35 publically available species of *Legionella* (Burstein *et al*, 2016). Hierarchical clustering was next used to group each protein based on correlations of inter-protein distance (Fig 6G). We observe clear

linkage between the phylogenies of MavQ and SidP, which is consistent with our model that the physical constraints imposed by direct effector–effector interplay can serve as potent modulators of pathogen evolution.

## Lem14 synergizes with SidP to inhibit yeast growth

In addition to capturing effector–effector suppression, our assay was also designed to detect synergy between effectors—in which

two effectors inhibit yeast growth in concert. Two surprises from the screen were that only one set of effectors specifically synergized in a one-to-one fashion and that this synergy involved SidP, the bifunctional effector/metaeffector described above. In addition to its inhibitory effects on MavQ, SidP synergizes with another effector, Lem14 (Lpg1851), to cause a severe growth defect when co-expressed in our yeast screen (Fig 7A and B). Strikingly, neither protein confers a measurable growth defect on its own (Fig 7B).

While revealing the full mechanistic basis of SidP-Lem14 synergy remained outside of the scope of this current study, the uniqueness of this pair and its linkage to our earlier work warranted at least some investigation. In contrast to the SidP-MavQ interaction described above, the synergistic interaction between SidP and Lem14 does not appear to involve a physical interaction as tested by the Y2H assay (Appendix Fig S8) and LUMIER (Table EV5). Providing further support to an indirect synergistic interaction, we observe clear cross-species synergy between *L. pneumophila* Lem14 and SidP from *L. dumoffii* in our screen (Fig 7A).

In notable contrast with SidP-MavQ suppression, the synergistic interaction between SidP and Lem14 requires both the active and full-length SidP protein (Fig 7B). To explore this further, we solved the crystal structure of Lem14, revealing a small, alpha-helical protein that forms an anti-parallel dimer (Fig 7C) and is structurally similar to another effector, LpiR1, whose molecular function remains elusive (Beyrakhova *et al*, 2016). Lem14 has a positively charged pocket that is required for SidP-Lem14 synergy in yeast, as mutants that reverse the charge (Fig 7C, inset) alleviate the SidP-Lem14 yeast growth inhibition (Fig 7B). Indeed, together with our suppression data, the observation of SidP-Lem14 synergy makes SidP a remarkable hub of effector activity within the eukaryotic cell, likely linking SidP/MavQ host lipid modulation with the as-of-yet undiscovered function of Lem14.

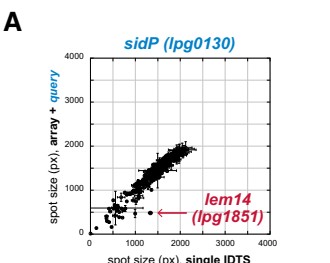
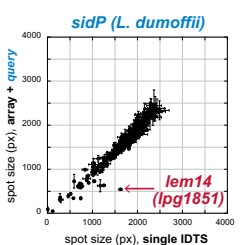

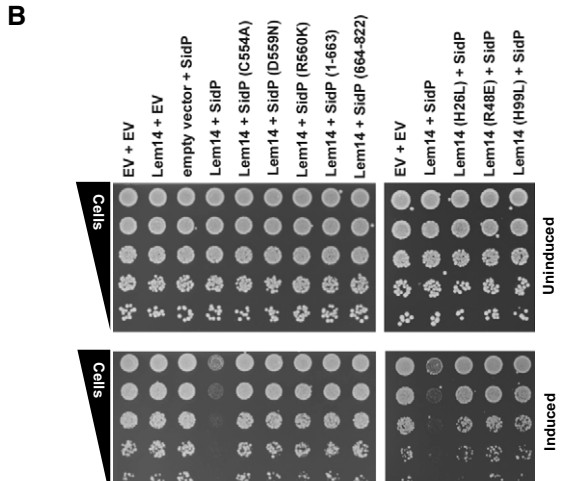

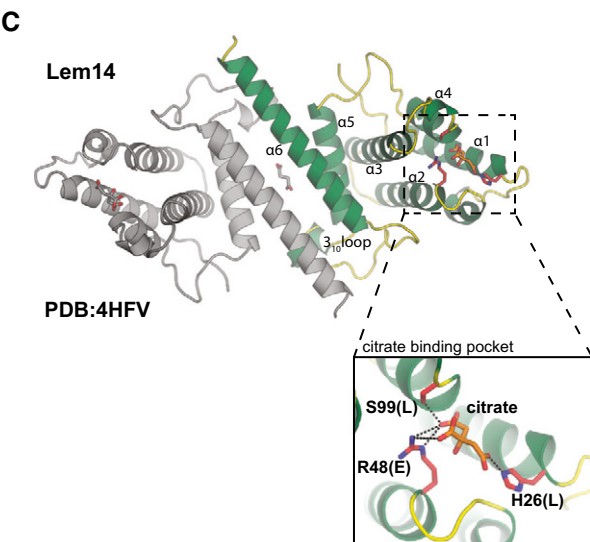

**Figure 7. SidP and Lem14 display an aggravating interaction.**

A  Effector activation between Lem14 and SidP revealed by the effector–effector interaction screen. Each of the IDTS in the array was mated to create either query-IDTS-containing diploids (*y*-axis) or vector-IDTS controls (*x*-axis). IDTS pairs that combine to cause a growth defect in yeast fall below the diagonal, as is the Lem14 array strain indicated in red. SidP and Lem14 are the only IDTS that display such a specific one-to-one phenotype in the screen. Querying the same array with the *L. dumoffii* ortholog of SidP also results in synergy with *L. pneumophila* Lem14 on the array; this cross-species behavior may reflect an indirect functional interaction between the two proteins (see Fig 6). Error bars represent the SD of spot size between quadruplicate spots within each dataset.

B  The synergistic effect of SidP and Lem14 is dependent on the SidP phosphatase activity and Lem14's charged pocket. Yeast strains co-expressing (i) SidP phosphatase mutants and fragments with wild-type Lem14 and (ii) co-expressing wild-type SidP with Lem14 mutants were tested in a yeast spot dilution assay as described previously. Co-overexpression of wild-type SidP and Lem14 caused a sicker than expected growth phenotype in yeast, but any SidP or Lem14 mutant alleviated this effect. Expression and stability of all mutants were tested (Appendix Figs S3A and S8B).

C  The 1.9 Å crystal structure of Lem14 (4HFV) shows it to be a small, predominantly alpha-helical protein forming an anti-parallel dimer, interfacing via head-to-tail α6-α6 hydrophobic interactions and interactions between the N-terminal portion of α5 and the 3₁₀-containing loop inserted between α5 and α6 of the neighboring chain. A citric acid molecule was found inside a positively charged pocket at the ends of α1–4.

# Discussion

Our systematic pairwise effector–effector interaction screen has highlighted the complex hierarchy present in the effector arsenal of *Legionella*. We captured all previously known examples of effector–effector suppression and identified an additional seventeen effector–effector suppression pairs, including nine novel putative metaeffectors (Fig 8A). While the concept of "metaeffector" remains fluid, there is a clear need to distinguish this special class of proteins from equally informative effector–effector relationships like that of SidM and SidD, where counteracting activities on a shared target reverse the effects of each effector on the host cell. We previously proposed that a defining characteristic of a metaeffector should be a direct physical interaction between it and its cognate IDTS, following the established relationship between canonical effectors and their host targets (Ensminger, 2016).

Our functional characterization of several metaeffectors identified in this study provides a number of important insights: we demonstrate that a single protein can function as both a metaeffector (targeting other IDTS) and a canonical effector (targeting the host) [SidP], these interactions may place evolutionary constraints on the plasticity of effector arsenals, and mechanisms of effector–effector suppression are diverse (Fig 8B). In light of these findings, we expect that as additional mechanisms of metaeffector function are discovered, they are likely to derive from as broad a set of activities as classical effectors that target the host.

It was recently shown that the cognate effector of LupA, LegC3, interferes with vesicle trafficking (de Felipe *et al*, 2008) by inhibiting the formation of endogenous trans-SNARE complexes during vacuolar fusion (Bennett *et al*, 2013) and forms a stable and functional SNARE acceptor complexes with two other IDTS YlfA/LegC7 and YlfB/LegC2 and the mammalian R-SNARE VAMP4 (Shi *et al*, 2016). In light of our data showing that YlfA is also directly inactivated by another novel metaeffector, MavE (Figs 1B and 2A), one attractive model is that LupA and MavE act to prevent detrimental off-target effects of their individual cognate effectors prior to complete SNARE complex formation. Indeed, such a model would explain why we observe yeast growth inhibition when we express either YlfA or LegC3 individually—and would reveal another important role for metaeffectors during infection.

By examining previously generated RNA-seq data for *Legionella pneumophila* str. Philadelphia-1 during intracellular replication in *Acanthamoeba castellanii* (Weissenmayer *et al*, 2011), we can begin to place several effector–effector interactions within the context of infection. Like the previously described SidH-LubX interaction, four additional pairs exhibit differential expression patterns consistent with the cognate effector being held in check during the replicative (mid) stages of infection: LegA8(AnkX)-Lem3, LegC3-LupA, SdbB-SdbC, and SidM(DrrA)-SidD (Appendix Fig S9A). Four other effector–effector pairs exhibit differential expression patterns consistent with the cognate effector being held in check during the transmissive (early/late) stages of infection: Ceg3-Lpg0081, SidE-SdjA|

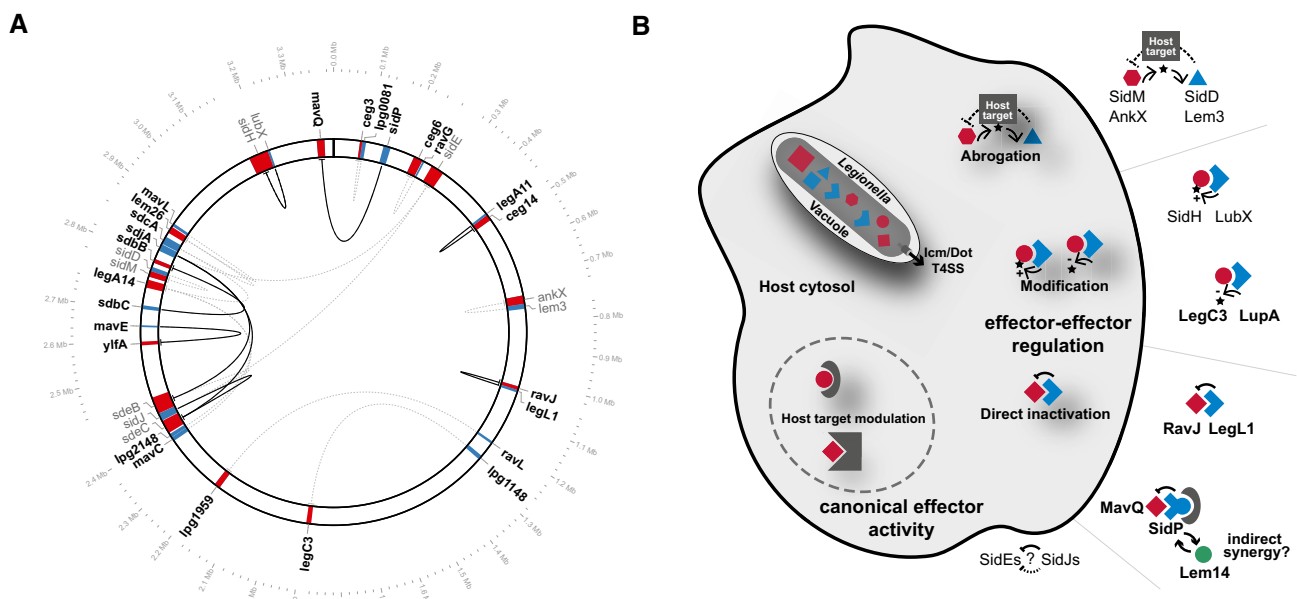

**Figure 8. A summary of effector–effector interactions revealed by our screen.**

A   Summary of all effector–effector suppression pairs and physical interaction data. Suppressing effectors (blue) and their cognate effectors (red) are shown at their genomic location. Previously identified effector–effector suppression pairs are shown in gray, and effector–effector suppression pairs identified in this study are labeled in black. Lines connect each pair (black: evidence for a physical interaction; dashed, gray: no evidence for a physical interaction). Excluding the multiple SidE/SidJ paralogs in the dataset, of the remaining effector–effector pairs six are immediately adjacent to one another on the chromosome, two are nearby (within 2–3 loci), and six are unlinked. The plot was generated using Circos v.0.69 (Krzywinski *et al*, 2009) with a 10× zoom at the effector genes shown.

B   Several mechanisms of effector–effector suppression. After release into the host cell, translocated bacterial substrates (effectors) regulate one another through several different functional interactions: indirectly, through counteracting modification of a shared host target, or directly through either steric complex formation or direct modification of one effector by another.

SidJ, Lpg1959-RavL, and LegA14-MavC|Lpg2148|SdcA (Appendix Fig S9B). Notably, the rest of our pairs do not display differential transcriptional profiles across the infection stage (Appendix Fig S9C). For these, we propose other mechanisms of regulation, including post-transcriptional regulation of protein levels, conditional-specific regulation, chaperone-based differential regulation of translocation to the host, or spatial regulation within the host cell in which effector activity is inhibited in only specific subcellular compartments. Indeed, a simple diffusion model could set up a spatial gradient of effector activity: close to the site of translocation, where effector and metaeffector concentration are highest, the two proteins would be more likely to physically interact. In this model, inactivation would predominate near the *Legionella*-containing vacuole; as effector and metaeffector diffuse further into the host cell, the frequency of effector–metaeffector physical interaction would reduce, providing an opportunity for unchecked effector activity to dominate. To fully explore the regulatory network of effectors and metaeffectors during infection, detailed proteomic analysis of effector protein levels and localization within the host cell are obvious next steps for the field. Notably, the mechanisms of inhibition we describe are likely to be critically important for interpreting these results, as metaeffectors such as LegL1 that rely upon steric hindrance of their cognate effectors will require absolute protein levels greater than catalytic antagonists such as LupA.

Current models of *Legionella* pathogenesis tend to view the pathogen's remarkable number of IDTS as a redundant cohort in support of its broad natural host range (Ensminger, 2016). Direct and indirect effector–effector suppression may provide a further explanation for IDTS expansion—as a mechanism to provide regulatory complexity to the progression of intracellular replication (Jeong *et al*, 2015; Ensminger, 2016). Indeed, some effector–effector regulation is likely a common feature of intracellular pathogens, reflecting the delicate balance that must be maintained between host perturbation and homeostasis for the duration of each replicative cycle. Effector–effector interaction may also provide a mechanism to regulate the assembly or disassembly of multi-effector complexes, or to compensate for a potentially leaky translocation system (Ensminger, 2016). To that end, the systematic identification of effector–effector interaction described herein should be extended to other pathogens of plants, animals, and humans—with the level of observed regulatory complexity providing key insights into how each pathogen balances the establishment and maintenance of its replicative niche.

## Materials and Methods

### *Saccharomyces cerevisiae* Icm/Dot-translocated substrate (IDTS) overexpression strains

330 pDONR221-IDTS constructs from a pDONR-IDTS library encompassing the Icm/Dot-translocated substrates first described by Losick *et al* (2010) and additional unpublished constructs (a kind gift from Ralph Isberg; see Table EV1) were used to clone IDTS into various yeast Gateway expression vectors (Alberti *et al*, 2007). To allow for expression of IDTS with an alternate start codon and to allow for assessment of expression of each IDTS by western blot, an N-terminal 1×HA-tag was introduced into pAG423GAL-ccdB and pAG416GAL-ccdB (Alberti *et al*, 2007). To make pAG423GAL-HA-ccdB

and pAG416GAL-HA-ccdB, pre-annealed oligos HA_pAGXXXgal_F (/5Phos/ctagtcttaccatgggttcttacccatacgatgttccagattacgcta) and HA_pAGXXXgal_R (/5Phos/ctagtagcgtaatctggaacatcgtatgggtaagaacccatggtaaga) were ligated into the SpeI site of pAG423GAL-ccdB and pAG416GAL-ccdB and the resulting clones were sequence-verified.

To create the BY4741+pAG423GAL-HA array strains, the pDONR-IDTS collection was cloned into the high-copy yeast expression vector pAG423GAL-HA-ccdB (GAL1 promoter, N-terminal HA-tag, and *HIS3* selectable marker) using LR clonase II (Life Technologies) according to the manufacturer's instructions. pAG423GAL-HA-IDTS clones were verified by PCR and transformed to the *S. cerevisiae* strain BY4741 (*MATa his3Δ1 leu2Δ0 lys2Δ0 ura3Δ0*; Brachmann *et al*, 1998) using the high-efficiency 96-well PEG/LiAc method (Gietz & Schiestl, 2007b).

The previously published *L. pneumophila* yeast expression library (Heidtman *et al*, 2009) in the high-copy vector pYES2 NT/A (Life Technologies, GAL1 promoter, N-terminal 6xHIS/Xpress tag, and *URA3* selectable marker) in the *S. cerevisiae* strain BY4742 (*MATα his3Δ1 leu2Δ0 met15Δ0 ura3Δ0*; Brachmann *et al*, 1998; Table EV6, Appendix Fig S1B) was extended by yeast homologous recombination cloning or ligation cloning to a total of 333 IDTS or putative IDTS (see Table EV7 for primers and cloning methods). Yeast homologous recombination cloning into pYES2 NT/A was performed as described previously (Heidtman *et al*, 2009) with the following modification: primers were designed with a 40-nt overlap with the vector sequence flanking the EcoRI-XhoI sites in the pYES2 NT/A multiple cloning site. Briefly, the IDTS ORFs were PCR-amplified from *L. pneumophila* str. Philadelphia-1 genomic DNA [GenBank Accession AE017354 (Chien *et al*, 2004; Rao *et al*, 2013)]. The resulting PCR products were transformed to BY4742 with EcoRI-XhoI-digested pYES2 NT/A using the high-efficiency PEG/LiAC method (Gietz & Schiestl, 2007a) and transformants were screened by PCR and sequence-verified. Previously, six Icm/Dot-translocated substrates (*lpg1368*, *lpg1488*, *lpg1489*, *lpg2157*, *lpg2504*, and *lpg2862*) could not be cloned by yeast homologous recombination and were designated non-recombinant (NR; Heidtman *et al*, 2009). In our extension of the pYES2 NT/A collection, we identified an additional eight NR IDTS (*lpg0090*, *lpg0693*, *lpg0234*, *lpg2153*, *lpg2461*, *lpg2519*, *lpg2523*, and *lpg2828*). Since the inability to clone these NR IDTS ORFs into a high-copy yeast expression vector is likely due to a combination of low-level leaky expression and their high toxicity to yeast (Heidtman *et al*, 2009), they and five other IDTS that failed to PCR-amplify or could not be screened in a high-copy plasmid due to phenotype (see Table EV7) were cloned into the low-copy pAG416GAL-HA-ccdB vector (GAL1 promoter, N-terminal HA-tag, *URA3* selectable marker) from the pDONR-IDTS collection using LR clonase II (Life Technologies) according to the manufacturer's instructions. These galactose-inducible low-copy plasmids were transformed to BY4742 using the high-efficiency PEG/LiAC method (Gietz & Schiestl, 2007a).

### Analysis of yeast growth defects

Liquid growth assays were performed in flat-bottom, clear 96-well plates (Greiner) sealed with adhesive plate seals (AB-0580, AB-gene) using a custom platform incorporating Tecan GENios plate readers (Tecan; Proctor *et al*, 2011). Freshly transformed BY4741 strains with pAG423GAL-HA-IDTS were transferred to a 96-well

plate with 100 µl SD-his + 2% glucose using a 96-floating pin tool (V&P Scientific) and grown overnight at 30°C. A fresh 96-well plate with 100 µl SD-his + 2% glucose or with 2% galactose was inoculated from the preculture plate using a 2-µl pin tool (V&P Scientific) and grown at 30°C with continuous shaking. Yeast growth was monitored up to 20 h by measuring the $OD_{595}$ every 15 min. The growth fitness of each strain was calculated as the ratio of the area under the curve (AUC) of a IDTS-expressing strain over a pAG423GAL-HA-ccdB empty vector control after 20 h. The average AUC ratio and standard deviation of three independent replicates were calculated.

The BY4741 with pAG423GAL-HA-IDTS collection and the 126 strains of the original BY4742 with pYES2 NT/A-IDTS collection (Heidtman *et al*, 2009) were arrayed in 384-well format with the border and empty spots filled with empty vector control. The 384-well format arrays were pinned in quadruplicate (1,536-density) onto SD-his + 2% galactose or SD-ura + 2% galactose, respectively, using the ROTOR HDA pinning robot (Singer Instruments) and grown for 2 days at 30°C. The arrays were then imaged using a high-resolution camera, and the spot sizes were quantified using SGAtools (http://sgatools.ccbr.utoronto.ca/; Wagih *et al*, 2013). Outlier spot sizes flagged by the Jackknife filter (JK) in SGAtools were removed, and the average and standard deviation of the remaining values were calculated and normalized to the average empty vector control (Tables EV1 and EV6, Appendix Fig S1). Five of the 330 array strains were filtered out after analysis because their identity could not be established, likely due to a mixture of clones within the spot on the array (*lpg1154*, *lpg0403*, *lpg0195*, *lpg2147*, and *lpg2806*). To estimate the total number of strains that have a growth defect in the BY4741 + pAG423GAL-HA-IDTS array, the empty vector spots of the inner rim of the border were averaged and compared to the average spot size of individual strains. In total, 227 strains have an (average strain spot size + stdDev) < (average empty vector spot size − StdDev).

### Suppression profiling screen

Each haploid query strain (BY4742 + pYES2 NT/A-IDTS) was individually mated to the BY4741 + pAG423GAL-HA strain array, similar to procedures in the SGA protocol (Tong & Boone, 2006), to test for possible rescue of yeast toxicity by each of the array overexpression strains. A 3-ml overnight culture of the IDTS query strains and an empty vector control query was spread onto a SD-ura + 2% glucose agar PlusPlates (Singer Instruments) and grown overnight at 30°C. The arrayed library (330 strains overexpressing individual IDTS and empty vector controls in 384-well density) was pinned in quadruplicate (1,536 density) onto SD-his + 2% glucose plate using the ROTOR HDA (Singer instruments) and 384 short pads (Singer Instruments) and grown overnight at 30°C. To mate the query with the array strains, the freshly grown query lawn was transferred to an YPD plate using the 1,536 short pads (Singer Instruments) and the 1,536-format array was pinned on top of the query spots. The strains were allowed to mate overnight at room temperature and subsequently pinned to media selecting for the presence of both plasmids (SD-ura/his + 2% glucose) to recover diploid strains. The diploid plate was grown for 1 day at 30°C and transferred to SD-ura/his + 2% galactose to induce expression of the query and array IDTS plasmids. These galactose plates were incubated at 30°C and imaged

after 1, 2, and 3 days using a high-resolution camera. Spot sizes were quantified using SGAtools (http://sgatools.ccbr.utoronto.ca/; Wagih *et al*, 2013), with the advanced option "keep large replicates". The SGA output files used in our analysis—all query strains (at 2 days)—will be deposited into the Dryad Digital Repository.

In 16 suppression profiling screens, the growth inhibition caused by the query IDTS was rescued by the overexpression of IDTS on the array (Fig 1B). These screens were subsequently repeated to confirm rescue. The two replicate screens were then analyzed as follows: outlier spot sizes flagged by the Jackknife filter (JK) in SGAtools were removed and the average and standard deviation of the remaining values were calculated. (Five array strains were filtered out after analysis because their identity could not be established, as described above.) For the strains that suppressed toxicity of the query IDTS in both replicates, the plasmid was recovered and sequenced to confirm identity. Converse pairwise suppression where the yeast growth inhibition was suppressed by a query IDTS was confirmed by yeast spot dilution assay (see below). Three colonies from the array IDTS were sequenced to confirm identity, mated with the suppressing query IDTS or empty vector control, and the resulting diploid strains were used in the spot dilution assay (Appendix Fig S2).

### Yeast two-hybrid assays

To investigate a possible physical interaction between the two partners of the effector–effector suppression pairs, we used the Y2H assay (Dreze *et al*, 2010). The strain Y8800 (*MATa leu2-3,112 trp1-901 his3-200 ura3-52 gal4Δ gal80Δ GAL2-ADE2 LYS2::GAL1-HIS3 MET2::GAL7-lacZ cyh2R*; Yu *et al*, 2008) and vectors with the DNA-binding (DB) and transcription-activating (AD) domain of Gal4 (pDEST-DB, pDEST-AD; Dreze *et al*, 2010) were a kind gift from N. Yachie and F. Roth (University of Toronto, Canada).

The suppressing IDTSs were cloned into Gateway destination plasmid pDEST-DB (ADH1 promoter, CEN, *LEU2* selectable marker, N-terminal Gal4 DNA-binding domain) by Gateway LR reaction using LR clonase II, transformed to Y8800 using the LiAc/PEG method (Gietz & Schiestl, 2007a), and plated onto SD-leu + ade/2% glucose. The IDTSs with a yeast growth defect were cloned into pDEST-AD (ADH1 promoter, CEN, *TRP1* selectable marker, N-terminal Gal4 transcription activation domain) by Gateway LR reaction using LR clonase II. Because the ADH1 promoter is a constitutive promoter, the AD-IDTS fusions or empty vector control was transformed to Y8800 + pDEST-DB-antagonist IDTS, to ameliorate the toxicity of the pDEST-AD-IDTS, and plated onto SD-leu/trp + ade/2% glucose. A single colony of Y8800 + pDEST-DB-IDTS/pDEST-AD-IDTS or pDEST-AD-empty vector strains was inoculated in a 96-well plate with 100 µl SD −leu/trp + ade/2% glucose medium, grown overnight and back-diluted 1:25 or 1:50 in fresh medium before spotting onto SD −leu/trp + ade/2% glucose and SD-leu/trp/his, +ade/2% glucose with or without 1 mM 3-amino-1,2,4-triazole (3-AT; Bioshop) using the VP 407AH pin tool. The plates were grown for 2 or 3 days before imaging. Positive Y2H growth (growth on −his and −his + 1 mM 3AT conditions) was screened for cryptic autoactivator mutations in the pDEST-DB plasmids by counterselecting for the pDEST-AD on SD-leu/his + ade/2% glucose and 1 mg/l cycloheximide as described (Dreze *et al*, 2010). All experiments were performed in triplicate.

## LUMIER with bait concentration assay

The functional antagonist IDTSs were cloned into the Gateway destination vectors pcDNA3.1-3XFLAG-V5-ccdB and pcDNA3.1-Renilla-ccdB (Taipale *et al*, 2012) from pDONR221 clones using Gateway technology. The LUMIER with BACON assay was performed as described previously (Taipale *et al*, 2012) with minor modifications. HEK293T cells were seeded into a 96-well plate at 30,000 cells/well density in 100 μl DMEM supplemented with 10% fetal bovine serum, penicillin/streptomycin, and L-glutamine and allowed to grow for 24 h before adding 50 μl OptiMEM (Life Technologies) with 100 μg/ml Lipofectamine 2000 (Life Technologies), 75 ng pcDNA3.1-FLAG-V5 bait, and 75 ng pcDNA3.1-Renilla-prey DNA. After 48 h, the cells were washed with 1× PBS and lysed in cold HENG buffer [20 mM Hepes-KOH pH 7.9, 150 mM NaCl, 2 mM EDTA pH 8.0, 0.5% Triton X-100, 5% glycerol, 1× EDTA-free complete mini protease inhibitor cocktail (Roche)]. The lysate was transferred to white 384-well LUMITRAC 600 high-binding plates (Greiner) coated with anti-FLAG-M2 (Cat# F1804, Sigma-Aldrich) and incubated for 3 h at 4°C. The 384-well plates were washed with cold HENG buffer and with cold 0.5 M NaCl/1% Triton X-100 HENG buffer before adding 2.5 μM coelenterazine h (Nanolight Technology) in Nanoluc assay buffer (20 mM Tris–HCl pH 7.5, 1 mM EDTA, 150 mM KCl, 0.5% Tergitol NP9). The luminescence signal was measured in a Synergy Neo plate reader (Biotec). The individual 3XFLAG-V5-bait concentration was determined by adding anti-FLAG-M2-HRP (Cat# A8592, Sigma-Aldrich). After 1 h incubation at room temperature, the plates were washed with PBS 0.05% Tween and the bait ELISA signal was measured using fivefold diluted Pico substrate (Thermo Scientific) in the above-mentioned plate reader.

The LUMIER interaction scores were calculated as Z-scores and filtered for detectable bait concentrations (Taipale *et al*, 2014). For each *Renilla* luciferase prey, luminescence signals were mean-normalized and log2-transformed and Z-scores were calculated from the estimated mean and standard deviation for all datapoints (Table EV5). Bait ELISA signals below the mean plus 2× standard deviation of ELISA signal from no-bait controls were removed from the final data set and are indicated as NA in Table EV5. The heatmap was generated using heatmap.2 without clustering in gplots v2.15.0 in R v3.1.1.

## Yeast spot dilution assay

The growth defect of MavQ (Lpg2975), RavJ (Lpg0944) mutants, the rescue of MavQ by SidP (Lpg0130) mutants, and the synergy between SidP and Lem14 mutants were assayed by a yeast spot dilution assay. Overnight cultures were grown in appropriate selective medium with 2% glucose, normalized to $OD_{600}$ of 1, and used to make dilution series with fivefold dilution steps. The dilutions were spotted onto selective media plates with 2% glucose or 2% galactose to induce expression, using the VP 407AH pin tool (V&P Scientific), and imaged after incubation for 2 days at 30°C.

To verify expression and stability of the MavQ, SidP, RavJ, and Lem14 mutants, strains were grown overnight in the SD-his + 2% glucose, back-diluted to $OD_{600} = 1.5$ in SD-his + 2% galactose, and incubated at 30°C for 5 h. Six $OD_{600}$ units of cells were harvested and lysed as described previously (von der Haar, 2007). Briefly, per three $OD_{600}$ units 200 μl lysis buffer (0.1 M NaOH, 2% SDS, 2%

β-mercaptoethanol, 50 mM EDTA) is added and incubated at 90°C for 10 min followed by the addition of 5 μl 4 M acetic acid and a second incubation at 90°C for 10 min. 50 μl of loading buffer (0.05% bromophenol blue, 250 mM Tris–HCl pH 6.8, 50% glycerol) was added. The equivalent of 0.4 $OD_{600}$ units of sample (RavJ and Lem14 mutants) or 0.8 $OD_{600}$ units (SidP and MavQ mutants) was analyzed by SDS–PAGE and western blot using anti-HA.11 (Cat# 901501, Clone 16B12, BioLegend Inc) or anti-Xpress (R910-25, ThermoFisher Scientific) antibodies.

## Protein purification

Gene fragments corresponding to proteins RavJ (Lpg0944) residues 1–230, 230–391, LegL1 (Lpg0945) 1–296, LupA (*Legionella* ubiquitin-specific protease A, Lpg1148) 1–373, SidP (Lpg0130), MavQ (Lpg2975) 1–871, and Lem14 (Lpg1851) 1–220 were PCR-amplified from *Legionella pneumophila* str. Philadelphia-1 genomic DNA and inserted into either plasmids p15TV-LIC (Eschenfeldt *et al*, 2009) or pET28-SBP-TEV (Addgene plasmid #36943), providing N-terminal 6xHIS-TEV or 6xHIS-SBP-TEV epitope tags, respectively. Additional point mutants were prepared by QuikChange mutagenesis (Stratagene). Plasmids were sequenced and subsequently transformed to either BL21 Gold or BL21(RIL)DE3 *E. coli* for purification.

For selenomethionine enriched proteins, cultures for each of the proteins were grown in M9 SeMET High-Yield growth medium (Shanghai Medicilon Inc.) at 37°C with shaking to an $OD_{595}$ of 1.2 followed by a reduction in the temperature to 16°C and overnight induction of protein expression using 0.4 mM IPTG. For native proteins, cultures were grown in LB and expression was induced at an $OD_{595}$ of 0.8 with 0.4 mM IPTG, with the exception of MavQ and MavQ D147A, which were grown in Studier auto-induction medium ZYM-5052 (Studier, 2005) at 20°C overnight. Cells were harvested by centrifugation at 9,300× *g*, resuspended in 50 mM HEPES pH 7.5, 500 mM NaCl, 5% glycerol, 5 mM imidazole, and lysed by sonication. Lysates were clarified by centrifugation at 21,000× *g* at 4°C and the supernatants incubated with Ni-NTA agarose (Qiagen) at 4°C with gentle mixing for 4 h, followed by washing with 50 mM HEPES pH 7.5, 500 mM NaCl, 5% glycerol, 30 mM imidazole. Protein was eluted using 50 mM HEPES pH 7.5, 500 mM NaCl, 5% glycerol, 250 mM imidazole. Cleavage of the His-tag prior to crystallization was performed by incubation of the purified protein with TEV protease followed by dialysis into 10 mM HEPES pH 7.5, 500 mM NaCl and finally removal of both TEV protease and the 6xHis-tag by further incubation with Ni-NTA agarose. Where required, proteins were further purified by size-exclusion chromatography on Superdex 200 HiLoad 16/60 column (GE Healthcare Life Sciences) in 0.5 M NaCl, 10 mM HEPES pH 7.5, after which the proteins were concentrated to approximately 10 mg/ml using a centrifugal concentrator (Corning) with an appropriate molecular weight cutoff and either used immediately or flash-frozen in liquid nitrogen for storage at −80°C.

## Crystallization and structure determination

For Lem14 structure determination, selenomethionine-substituted protein was crystalized at room temperature by hanging-drop vapor diffusion, the final crystallization solution contained 0.2 M

diammonium citrate, 20% PEG3350, 1/70 thermolysin, pH 5.0. For the LupA (1–373) structure determination, selenomethionine-substituted protein was crystallized at room temperature by hanging-drop vapor-diffusion method and the successful condition contained 10 mM $MgCl_2$, 4% sucrose and 1.6 M $NH_4$ sulfate. For the RavJ (1–230) structure determination, native protein was crystalized at room temperature by sitting-drop vapor diffusion and the final crystallization solution contained 1.6 M ammonium sulfate, 0.1 M NaCl, 0.1 M HEPES pH 7.5. For the RavJ (230–391) structure determination, selenomethionine-substituted protein was crystalized at room temperature by hanging-drop vapor diffusion and the final crystallization solution contained 1.6 M ammonium sulfate, 0.1 M NaCl, 0.1 M HEPES pH 7.5. Finally, for the RavJ (1–230 fragment)-LegL1 complex structure, the selenomethionine-containing protein sample of the RavJ (1-230 fragment) was pre-incubated with LegL1 native protein sample in 1:1 ratio for 1 h at 4°C in 0.1 M NaCl and 10 mM HEPES pH 7.5 buffer. This protein mixture was run through a Superdex 200 HiLoad 16/60 column (GE Healthcare Life Sciences) calibrated in the same buffer, and the fraction corresponding to the RavJ (1–230)-LegL1 complex was collected and concentrated to 57 mg/ml. Diffraction quality crystals were obtained by mixing 0.5 µl of this protein complex sample in 1:1 ratio with crystallization solution containing 0.1 M NaCl, 100 mM Tris pH 8.5 and 26% (w/v) PEG3350 by hanging-drop vapor-diffusion method at room temperature. All crystals were cryo-protected by immersion in paratone-N oil before being flash-frozen in liquid nitrogen.

The X-ray diffraction data were collected at the 19-BM and 19-ID beamlines at the Structural Biology Center, Advanced Photon Source, Argonne National Laboratory, at the selenium absorption peak wavelength at 100 K (Rosenbaum et al, 2006). All datasets were processed with HKL-3000 (Minor et al, 2006). The structures of RavJ (1–230), RavJ (230–391), the RavJ (1–230)—LegL1 complex and of LupA were determined using the SAD phasing method, and the structure of Lem14 was determined using the MAD phasing method using the program Shelx (Sheldrick, 2010). Model building was initially performed by ShelxE, and the initial models were further extended using the ARP/wARP web service (Langer et al, 2008), followed by manual adjustments. All structures were refined using the program Phenix.refine (Adams et al, 2010) and manually inspected and rebuilt using the program Coot (Emsley & Cowtan, 2004). Translation–libration–screw rotation (TLS) parameterization as defined by the TLSMD server (Painter & Merritt, 2006) was used for refinement of the complex. Geometric suitability for all structures was verified using the Phenix valida-tion tools and the wwPDB validation server. The data collection and refinement statistics are summarized in Table EV4. Structural comparisons were performed using the program COFACTOR (Roy et al, 2012).

## Homology modeling of SidP (Lpg0130)

Homology modeling of Lpg0130 was performed using Phyre2 (Kelley & Sternberg, 2009) in intensive mode. The resulting model was based on a single template (LLO_3270, PDB: 4JZA; Toulabi et al, 2013, 53% identical) with a 100% confidence score. 99% of residues were modeled at > 90% confidence. Only 10 residues were modeled using ab initio methods.

## MavQ (Lpg2975) homology detection and HMM-HMM alignment

The MavQ amino acid sequence was submitted to HHPred (Soding, 2005) with suggested default parameters (HMMdatabase: pdb70_05Jun16, MSAgeneration HHblits, Max 3 iterations, secondary structure scoring, local alignment). The N-terminal part of MavQ may share some homology with several PI3 and PI4 kinases and aminoglycoside phosphotransferases [4h05, 4hne (Zhou et al, 2014), 4wtv (Klima et al, 2015), 3w0o (Iino et al, 2013), 4bfr (Certal et al, 2014), and 4ykn (Yang et al, 2015)]. The resulting HMM-HMM alignments with MavQ were used to assist in determining potentially important residues. Residues S25/26, K46, D147A, and D160A were mutated to alanine using site-directed mutagenesis.

## In vitro determination of MavQ ATP hydrolysis activity

ATP-to-ADP conversion by MavQ was performed according to Zhou et al (2014). Briefly, PI diC8 dissolved in water was dried in a vacuum centrifuge, resuspended to a concentration of 7 mM in kinase buffer (20 mM Tris pH 7.5, 150 mM NaCl, 0.2% Triton X-100, 1 mM EDTA, 20 mM $MgCl_2$), and sonicated until translucent. 0.5 µg of MavQ or MavQ D147A (1 mg/ml) in kinase buffer was added to 5 µl of resuspended lipid micelles with a final volume of 9 µl. A third reaction without PI with wild-type MavQ was also prepared. Reactions were started by the addition of 1 µl of 10 mM ultrapure ATP, incubated for 30 min at room temperature, and were performed in duplicate. ATP-to-ADP conversion was measured using ADP-Glo kinase assay kit (Promega) according to the manu-facturer's instructions. Luminescence was recorded using a Tecan plate reader with an integration time set to 750 ms.

## AP-MS analysis of MavQ-SidP, LegL1-RavJ, and RavJ-host interactions

Legionella Lp03 lysate from exponential and post-exponential cells for AP-MS analysis of Mav-SidP and LegL1-RavJ interactions was prepared as described (Quaile et al, 2015). U937 lysate for RavJ-host interactions was prepared from U937 cells grown in RPMI 1640 supplemented with 10% FBS and 1% penicillin/strepto-mycin. Cells were harvested by centrifugation (150× g, 5 min), washed once with PBS, and stored at −80°C. Per technical repli-cate, $1 \times 10^8$ frozen U937 cells were thawed in 1 ml of lysis buffer (50 mM HEPES pH 8.0, 150 mM NaCl, 5 mM EDTA, 5 mM DTT, 0.1% Nonidet P-40, 1× EDTA-free complete mini protease inhi-bitor cocktail (Roche)), lysed by five rounds of freeze/thaw in liquid nitrogen, and cleared by centrifugation (10 min, 16,000× g, 4°C). Lysate was depleted of endogenous biotin using 50 µl/ml washed streptavidin magnetic sepharose beads (New England Biolabs) for 45 min.

AP-MS analysis was performed as described (Quaile et al, 2015). Briefly, 50 µl of streptavidin mag sepharose beads with 6XHis-SBP tagged RavJ, LegL1, or MavQ were added to a 1-ml aliquot of clarified, biotin-depleted lysate (Lp03 or U937) and incubated with gentle mixing at 4°C for 3 h. Beads were washed twice with 1 ml lysis buffer, transferred to a fresh tube followed by a final wash with 500 µl lysis buffer. Bait and bound proteins were eluted with 100 µl of elution buffer (50 mM ammonium bicarbonate, 2.5 mM biotin) on ice for 10 min. One microgram of

sequencing grade modified trypsin (Promega) was added to the eluate, and proteins were digested overnight at 37°C. Digestion was terminated by adding trifluoroacetic acid (TFA) to a final concentration of 0.2% (v/v), and peptides were desalted and concentrated using Agilent C18 OMIX tips according to the manufacturer's instructions and dried to completion in a vacuum centrifuge. Samples were resuspended in 0.1% formic acid and analyzed by mass spectrometry on an LTQ XL mass spectrometer (Thermo Scientific). Raw instrument data were converted to mzXML format using msconvert (Holman *et al*, 2014), and peptide spectra were searched using X! Tandem (Craig & Beavis, 2004). Protein/peptide identifications were further compared and analyzed using ProHits VM (Liu *et al*, 2012). A number of no-bait controls, as well as several other IDTSs, were used to identify non-specific binding proteins. The mass spectrometry proteomics data have been deposited to the ProteomeXchange Consortium (http://proteomecentral.proteomexchange.org) via the PRIDE partner repository (Vizcaino *et al*, 2013) and are summarized in Table EV3 (MavQ or LegL1 with Lp03 lysate) and Table EV8 (RavJ with U937 lysate).

### Lipid overlay blots with SidP, MavQ, and Lem14

6xHisSBP-tagged SidP, MavQ, and Lem14 were purified using the 6xHis-tag, as described above, followed by an additional SBP-tag purification step with streptavidin mag sepharose (GE Healthcare Life Sciences) and eluted in 50 mM HEPES pH 7.5, 300 mM NaCl, 2.5% glycerol, and 4 mM biotin. The lipid overlay blot was performed as described (Weber *et al*, 2013), with minor modifications. Briefly, the PIP strips (Echelon, P-6001) were blocked in 3% fatty acid-free BSA (Roche) in Tris-buffered saline with 0.1% Tween 20 (blocking buffer) and incubated overnight in the dark at 4°C with 5 pmol/ml purified protein in a total volume of 10 ml blocking buffer. The PIP strips were washed with blocking buffer, incubated with mouse anti-polyhis antibody (1:2,000, clone HIS1, Sigma-Aldrich) in blocking buffer, washed, and incubated with anti-mouse HRP antibody (7076, Cell Signaling Technologies) in blocking buffer.

### Gel filtration chromatography of the RavJ-LegL1 complex

One milligram of each protein was analyzed by gel filtration, in the combinations specified or alone using an AKTA Explorer 100 system (GE Healthcare Life Sciences) equipped with a Superdex 200 10/300 column (GE Healthcare Life Sciences). The column was equilibrated and run in 10 mM HEPES pH 7.5, 100 mM NaCl at a flow rate of 0.5 ml/min. Elution profiles were measured by absorbance at 280 nm. $A_{280}$ data from multiple runs were normalized to 100% and aligned according to elution time.

### *In vitro* determination of LupA (Lpg1148) hydrolase substrate specificity

To determine the hydrolase specificity of LupA, ubiquitin-like substrates (ubiquitin, SUMO, NEDD8) derivatized with 7-amido-4-methylcoumarin (AMC) on the carboxy-terminus were employed. All substrates were purchased from Boston Biochem (USA) and used as per manufacturer's recommendations. Briefly, the assay

was set up in a 96-well plate format (Microfluor1 plates, ThermoScientific) in a 200 µl reaction volume. For initial substrate specificity tests, the purified Lpg1148 1–373 fragment was diluted into the reaction buffer (50 mM HEPES pH 7.8, 50 mM NaCl, 0.5 mM EDTA, 1 mM DTT) to a final concentration of 50 nM, and the reaction was initiated with 5 µl of substrate resulting in the following final concentrations: 300 nM Ubiquitin-AMC, 190 nM SUMO-3-AMC, 180 nM SUMO-2-AMC, 120 nM SUMO-1-AMC, and 180 nM NEDD8-AMC. Substrate hydrolysis was measured as fluorescence of released AMC ($Ex_{380\ nm}$, $Em_{460\ nm}$). The reaction was monitored in continuous mode using the SpectraMax Plate Reader (Molecular Devices) for 1 h at 23°C. Reaction conditions used to assess the effects of LupA active site mutant H183A on the ubiquitin-AMC substrate were the same as described above, but was monitored for 1.5 h at 23°C. A reaction mixture containing no enzyme was used to monitor spontaneous degradation of the substrates. Potential non-specific proteolysis carried over from enzyme purification was monitored by using identical experimental conditions with an unrelated protein (Lpg0439) purified under the same conditions as LupA (1–373).

### *In vivo* deubiquitination of LegC3 by LupA (Lpg1148)

HEK293T cells were grown using standard techniques; incubated at 37°C, 5% $CO_2$ in DMEM (Gibco) supplemented with 10% FBS (Gibco), and 1% penicillin/streptomycin (Sigma-Aldrich). Cells were transfected with pRK5-HA-Ubiquitin (Addgene plasmid # 17608; Lim *et al*, 2005), pcDNA3.1-3XFLAG-V5-LegC3 (see LUMIER assay), and either pcDNA3.1-nV5-DEST (Life Technologies) empty vector, LupA, LupA C252A, H183A, or D207A mutants as indicated, at approximately 60–70% cell confluence using Lipofectamine 3000 (Life Technologies) according to the manufacturer's instructions. After 24 h, cells were washed once with PBS, harvested by centrifugation, and frozen at −80°C.

Denaturing IPs were performed as described (Tansey, 2007) with minor modifications. Briefly, cell pellets from a 10-cm plate were thawed into 200 µl TSD lysis buffer (50 mM Tris pH 7.5, 1% SDS, 5 mM DTT, 1× EDTA-free complete mini protease inhibitor cocktail (Roche)) and heated at 98°C for 10 min. Lysates were clarified by centrifugation (18,000× *g*, 5 min), and 100 µl of supernatant was diluted with 1.2 ml of TNN immunoprecipitation (IP) buffer (50 mM Tris pH 7.5, 250 mM NaCl, 5 mM EDTA, 0.5% Nonidet P-40). 130 µl of clarified lysate was retained for analysis of inputs. All subsequent incubation steps were performed at 4°C with gentle rotation. Lysates were precleared with 50 µl of washed protein G magnetic beads (NEB) to minimize non-specific binding. The precleared lysates (without beads) were transferred to a fresh tube and incubated with 5 µg of mouse anti-FLAG M2 antibody 1 h, after which 25 µl of protein G magnetic beads were added and incubated for 1 h. Beads were washed three times with IP buffer, resuspended in 30 µl 3× SDS–PAGE loading buffer, and heated at 98°C for 7 min.

26 µl of each IP was analyzed by SDS–PAGE and western blot and probed for HA-Ub using mouse anti-HA-7 (1:20,000, Cat# H3663, Sigma-Aldrich) in 1% BSA in PBS-T and TrueBlot ULTRA anti-mouse Ig HRP (1:2,000, Cat# 18-8817-31, Rockland Inc) in 5% milk, TBS-T. The western blot was stripped by three 10-min incubations in low pH glycine stripping buffer (0.1 M glycine, 20 mM

magnesium acetate, 50 mM KCl, pH 2.2) and reprobed for FLAG-V5-LegC3 using mouse anti-FLAG M2 (1:2,000, Cat# F1804, Sigma-Aldrich) in TBS-T and TrueBlot ULTRA anti-mouse Ig HRP as above. 10 μl of clarified input lysates with 5 μl of 3× SDS–PAGE loading dye was analyzed by SDS–PAGE and western blot using mouse anti-V5 antibody (1:2,000, Cat# R960, ThermoFisher Scientific) in 5% milk, TBS-T, and anti-mouse IgG HRP antibodies (1:2,000, Cat# 7076, Cell Signaling Technologies) in 5% milk, TBS-T.

### *Legionella dumoffii* ortholog identification and cross-species rescue

An Ontario *Legionella dumoffii* strain (*L. dumoffii* str Hamilton, a kind gift from C. Guyard and Public Health Ontario) was subjected to Illumina MiSeq 250x250x8 (paired-end) sequencing at the Donnelly Sequencing Centre, University of Toronto, as described previously (Rao *et al*, 2013). The resulting raw paired-end sequence reads were deposited as Illumina FASTQ files to the NCBI sequence read archive (study accession: SRP051121). Contigs were assembled from these reads using the Velvet *de novo* assembler (Zerbino & Birney, 2008) and subjected to automatic annotation using Prokka (Seemann, 2014) with default settings. Orthologs for the *L. pneumophila* genes *lpg2975*, *lpg0130*, *lpg0695*, and *lpg0696* were identified with PGAP (Zhao *et al*, 2012) using the MultiParanoid method with default settings. The *L. dumoffii* orthologs (Table EV9) were amplified by PCR from *L. dumoffii* str Hamilton genomic DNA and cloned by Gateway BP reactions with BP clonase (Life Technologies) to pDONR221-ccdB (Life Technologies). The resulting pDONR-IDTS plasmids were sequence-verified and the yeast toxic IDTS orthologs, *LdumoAE_01458* (*mavQ*) and *LdumoAE_01458* (*ankX*), were cloned into pAG426GAL-ccdB and pDEST-AD, while the cognate IDTS orthologs, *LdumoAE_01218* (*sidP*) and *LdumoAE_01010* (*lem3*), were cloned into pAG423GAL-HA-ccdB and pDEST-DB. Spot dilution assays and Y2H assays were performed as described above.

### MirrorTree analysis

To examine evolutionary relationships between conserved IDTSs, genome sequences of 41 *Legionella* species described by Burstein *et al* (2016) were downloaded from NCBI. Based on the assigned *Legionella* ortholog groups (LOGs; Burstein *et al*, 2016), 36 of the 41 species, excluding *L. adelaidensis*, *L. londiniensis*, *L. oakridgensis*, *L. maceachernii*, and *L. drancourtii*, show one single ortholog (no paralogs predicted) each of SidP (Lpg0130) and MavQ (Lpg2975). *Legionella cherrii* was also excluded from the analysis due to an assembly gap in the SidP ortholog gene. Based on the similar criterion of single existence in all the remaining 35 *Legionella* species, 16 other conserved IDTSs were selected. Protein sequences of each IDTS in the 35 *Legionella* species were aligned using the ClustalW option in MEGA v6.0. These alignments were then input to MirrorTree software (Pazos & Valencia, 2001; software kindly provided by Florencio Pazos). Results were represented as co-evolution correlation values between each two conserved IDTSs. Hierarchical clustering was then performed based on euclidean distance of correlation values. The heatmap was generated using heatmap.2 in gplots v2.15.0 in R v3.1.1.

The 35 species used in MirrorTree analysis are as follows: *L. pneumophila, L. longbeachae, L. anisa, L. birminghamensis, L. bozemanii, L. brunensis, L. cincinnatiensis, L. drozanskii, L. dumoffii, L. erythra,* *L. feeleii, L. geestiana, L. gormanii, L. gratiana, L. hackeliae, L. israelensis, L. jamestowniensis, L. jordanis, L. lansingensis, L. micdadei, L. moravica, L. nautarum, L. parisiensis, L. quateirensis, L. quinlivanii, L. rubrilucens, L. sainthelensi, L. santicrucis, L. shakespearei, L. spiritensis, L. steelei, L. steigerwaltii, L. tucsonensis, L. waltersii,* and *L. worsleiensis.*

### Data accessibility

Coordinates of Lem14 (4HFV), RavJ 1–225 (4RXV), RavJ 251–371 (4RXI), the complex structure of RavJ (1–230) and full-length LegL1 (4XA9), and LupA 123–403 (5DGG) have been deposited in the Protein Data Bank. SGA output files from the genetic interaction screen, including raw spot size information, have been deposited into the Dryad Digital Repository (doi: 10.5061/dryad.kj666).

**Expanded View** for this article is available online.

### Acknowledgements

We thank members of the Ensminger and Savchenko laboratories, including V. Cartier-Archambault for assistance with mining the published RNA-seq data. We thank J. Claycomb, L. Cowen, J. Brumell, C. Gradinaru, and M. Machner for critical comments; C. Nislow for scientific advice; M. Meneghini and C. Boone for equipment use; and F. Roth and N. Yachie for Y2H strains and plasmids. We thank all members of the Structural Biology Center at Argonne National Laboratory, especially A. Joachimiak, for help with protein crystallography experiments. This work was supported by an operating grant from the Canadian Institutes of Health Research (MOP-133406) to AWE, an infrastructure grant from the Canada Foundation for Innovation and the Ontario Research Fund (30364) to AWE; in part by a National Institutes of Health grant (GM094585) to AS through the Midwest Center for Structural Genomics and by the U. S. Department of Energy, Office of Biological and Environmental Research (contract DE-AC02-06CH11357); and by an operating grant from the Canadian Institutes of Health Research (MOP-142384) to MT.

### Author contributions

MLU, AWE, and AS conceived and designed the screen for effector–effector interplay. AWE, AS, and MLU with input from other authors prepared the manuscript. AWE, AS, and MLU supervised the project. MLU performed the screen and targeted assays for effector–effector interactions with library construction assistance from RDL and AWE. CO assisted MLU with cross-species analysis of effector–effector interactions. MLU performed the LUMIER assay with assistance from MT and ML. MM performed the *in vitro* assays of Lpg1148 deubiquitinase activity. ATQ performed the *in vivo* assays of this activity on LegC3, the assays on MavQ activity, and RavJ-host interaction identification. ATQ and EE performed analytical gel filtration analysis. CR performed MirrorTree analysis. PJS, MEC, JO, BPN, KM assisted in the determination of RavJ, RavJ/LegL1 complex, LupA, and Lem14 crystal structures.

### Conflict of interest

The authors declare that they have no conflict of interest.

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
