## [Review Process File · Molecular Systems Biology]

Diverse mechanisms of metaeffector activity in an intracellular bacterial pathogen, *Legionella pneumophila*.

Malene Urbanus, Andrew Quaile, Peter Stogios, Mariya Morar, Mr. Chitong Rao, Miss Rosa Di Leo, Elena Evdokimova, Ms. Mandy Lam, Ms. Christina Oatway, Marianne Cuff, Jerzy Osipiuk, Karolina Michalska, Boguslaw Nocek, Mikko Taipale, Alexei Savchenko and Alexander Ensminger

*Corresponding authors: Alexander Ensminger, University of Toronto;
Alexei Savchenko, University of Toronto*

Review timeline:	Submission date:	10 October 2016
	Editorial Decision:	21 October 2016
	Revision received:	13 November 2016
	Accepted:	17 November 2016

Editor: Maria Polychronidou

Transaction Report:

1st Editorial Decision

21 October 2016

Thank you again for submitting your work to Molecular Systems Biology. We have now heard back from two of the three referees who agreed to evaluate your manuscript. Since the recommendations of these two referees are rather similar, I prefer to make a decision now rather than further delaying the process. As you will see below, reviewers #2 and #3 acknowledge that the study seems very interesting. They raise however a series of (mostly minor) concerns, which should be carefully addressed in a revision of the manuscript.

REFeree REPORTS

Reviewer #2:

This is a nice piece of work of systematic identification of Legionella effector-effector interaction network. By screening yeast growth expressing effector-effector pairs, the authors successfully found all effector-effector interactions previously known, and further identified and characterized novel effector-effector interactions. This work further substantiates metaeffector (effector of effectors) hypothesis and shed light on how host-pathogen interaction is regulated by metaeffectors. I have several minor comments listed below.

1. I feel a little bit difficulty to grasp rationale of the screening and what panels of Figure 1B mean. It is worth to add a summary figure explaining screening strategy or sentences in the main text to briefly explain the outline of entire screening.
2. Remove underline from lines 148-149

3. "non-proteolytic (activating) ubiquitination" (line 199) sounds a bit strange.
4. "(Note that cross-species Y2H is technically not possible due to lack of cross-species rescue)" (lines 224-225).

I do not get the point of this sentence. It is technically possible to perform Y2H between SidPdumoffii/MavQpneumophila and SidPpneumophila/MavQdumoffii, and according to the authors' hypothesis these should result in negative data. Together with the dumoffii/dumoffii and pneumophila/pneumophila data presented in the manuscript, the experiment would strengthen the authors' hypothesis.

Reviewer #3:

The authors present a comprehensive screen in yeast of known secreted Legionella effectors. They concentrate on finding suppressive effector-effector interactions, and successfully recapitulate all such known modulations, as well as find novel and specific effector-effector suppression and one example of effector-effector synergy.

They go on to study some of these in detail and demonstrate both direct effector interactions, and indirect synergy. Interestingly some of these interactions share effector partners, suggesting that some effectors may be signalling/interaction hubs within complex networks.

In the main the paper is well-presented, and experiments are well-controlled and elegantly performed. The data largely supports the authors' conclusions and is both novel and extremely interesting. In addition it provides a resource for further investigation into effector regulation and interaction and will thus be of interest to many in the Legionella and wider bacterial pathogenesis community.

Specific points:

The introduction is mostly clear and straightforward, but I would suggest that the final sentence of paragraph 1 (p3 lines 55-57) replace the final sentence of the introduction.

In places throughout the text words have been italicised for emphasis - in most cases I think this is unnecessary since the text is largely clear and the novelty of the findings is obvious and need not be overstated.

P4-5 lines 95-98 - It is unclear to me why the authors describe the second part of their screen as being in the opposite direction to the first (p4, lines 95-98), since the readout is the same (suppression by a query effector of growth restriction caused by another effector). This should be clarified in the text.

P5 line 103 suggests that several effector pairs are genetically unlinked (presumably based upon their physical distance on the chromosome) - however it would be useful to examine if there are obvious shared regulatory/promoter motifs in these pairs, or if they are otherwise co-regulated. For example are the pairs that appear to be largely similar in transcriptional dynamics (Figure S9C) also those closely placed on the genome (and thus possibly part of the same operon)? A cursory examination of figure 8A shows this to be the apparent case, except for mavQ and sidP. This merits further analysis and discussion.

The RNAseq data in Figure S9 is mentioned in the discussion, I was unsure why this was not included more prominently in the results section. Also it is unclear as to what extent this supports the authors' suggestions that it shows "holding in check" of effector function. The data is all relative expression (on a per-gene basis), and so it is not apparent that even large relative changes represent absolute changes that might merit significant changes in post-transcriptional regulation. Of course my objection also presumes that RNA levels correlate with active, secreted effector levels outside of the bacteria, but this point also has some bearing upon the authors' interpretation.

P6 line 130/131 - It would be useful to describe a little of what is known about MavQ (or even merely to state it is an effector of unknown function).

P7 line 148/149 - Some of the text is underlined, reasons unclear.

Figure 5C - The authors describe the activity of the H183A mutant as being below the detection limit of the assay - they should show the detection limit on the graphs in 5B and C. Alternatively, if they mean that the results were not statistically different to no enzyme controls, then the appropriate statistical test should be described and p values (or other measure) shown on the graphs.

P10 line 222 - Cross-species rescue was not maintained for both gene pairs; this is correctly stated in line 225.

P10 line 241 - "Cross-species rescue" should be changed to "cross-species synergy", since no suppression/rescue occurs.

P10-11 lines 234-254 - This section of the paper feels quite preliminary and in contrast to much of the detailed analysis elsewhere it seems the authors are stretching towards a mechanism, but failing to demonstrate one. Whilst I will agree if the authors state that this mechanism is beyond the scope of the paper, I feel that this section feels like a slight anti-climax before moving into the discussion. However, the discovery of such a stark synergy is very interesting and so I do feel a slight re-emphasis of this section would help increase the readability of the paper.

P 18 Line 415 - "(C) A predicted catalytic residue (H183) abolished the ... activity" should read "(C) Mutation of a predicted ..."

The authors might consider swapping Figures 7B and C, since C is referred to in the text prior to B.

In Figure S9 lubX is referred to as legU2 - this should be changed to be consistent with the text. Similarly for legc7/yflA in figure 8A.

1st Revision - authors' response

13 November 2016

Our point-by-point response to the reviewers:

Reviewer #2:

This is a nice piece of work of systematic identification of Legionella effector-effector interaction network. By screening yeast growth expressing effector-effector pairs, the authors successfully found all effector-effector interactions previously known, and further identified and characterized novel effector-effector interactions. This work further substantiates metaeffector (effector of effectors) hypothesis and shed light on how hostpathogen interaction is regulated by metaeffectors. I have several minor comments listed below.

1. I feel a little bit difficulty to grasp rationale of the screening and what panels of Figure 1B mean. It is worth to add a summary figure explaining screening strategy or sentences in the main text to briefly explain the outline of entire screening.

Thank you for the suggestion. While figure 1A was meant to explain some of the logistics of the screen, it is indeed a bit complicated. The main text (Results, paragraph 2) has been expanded to more clearly explain the screening approach and all of its steps. Figure 1B's legend was modified slightly to explain that each axis is one of two biological replicates performed independently, which should help in its readability.

We have also updated Table EV7 to include the identity of all the clones present in the pYES2 NT/A library, not just the new clones we added to the existing, 127 clone Heidtman, 2009 collection.

Together, we hope these changes will make it easier for readers to get a clear understanding of library composition and the cloning approach.

2. Remove underline from lines 148-149

Thank you. The text was changed as noted.

3. "non-proteolytic (activating) ubiquitination" (line 199) sounds a bit strange.

Agreed. We meant to suggest a model in which the ubiquitination of LegC3 is not a mark for proteasomal degradation, but rather appears to positively contribute to LegC3's inhibition of yeast growth. We have reworded the phrase to clarify our language: "Lpg1148... removes a ubiquitin modification from LegC3 that otherwise supports its activity in a proteasomal-independent manner."

4. "(Note that cross-species Y2H is technically not possible due to lack of cross-species rescue)" (lines 224-225). I do not get the point of this sentence. It is technically possible to perform Y2H between SidPdumoffii/MavQpneumophila and SidPpneumophila/MavQdumoffii, and according to the authors' hypothesis these should result in negative data. Together with the dumoffii/dumoffii and pneumophila/pneumophila data presented in the manuscript, the experiment would strengthen the authors' hypothesis.

We apologize for the confusion. One limitation of the standard Y2H assay that is relevant to this potential application is that it depends on the expression of the HIS3 reporter and the resultant ability of strains to grow on histidine-deficient media. While our functional observations (lack of cross-species rescue) strongly suggest that we would not observe a physical interaction between MavQ and SidP orthologs from different species, the readout of such a result (no growth on HISmedia) is indistinguishable from the growth inhibition caused by MavQ in the absence of an effective suppressor. Other assays or modification of the assay might avoid these limitations, but their optimization is probably outside the scope of this study.

We have reworded the sentence to clarify the initial intent of our statement and to emphasize its tangential relationship to the bulk of the results:

“(A technical aside: the standard Y2H assay relies on expression of a growth-based reporter (*HIS3*) and resultant growth on histidine-deficient media, thus any examination of cross-species physical interaction using this approach would be obscured by MavQ's unchecked growth inhibition.)”

Reviewer #3:

The authors present a comprehensive screen in yeast of known secreted Legionella effectors. They concentrate on finding suppressive effector-effector interactions, and successfully recapitulate all such known modulations, as well as find novel and specific effector-effector suppression and one example of effector-effector synergy.

They go on to study some of these in detail and demonstrate both direct effector interactions, and indirect synergy. Interestingly some of these interactions share effector partners, suggesting that some effectors may be signalling/interaction hubs within complex networks.

In the main the paper is well-presented, and experiments are well-controlled and elegantly performed. The data largely supports the authors' conclusions and is both novel and extremely interesting. In addition it provides a resource for further investigation into effector regulation and interaction and will thus be of interest to many in the Legionella and wider bacterial pathogenesis community.

Specific points:

The introduction is mostly clear and straightforward, but I would suggest that the final sentence of paragraph 1 (p3 lines 55-57) replace the final sentence of the introduction.

We thank the reviewer for this suggestion. We completely agree that the suggested change improves upon the introduction and have adopted it.

In places throughout the text words have been italicised for emphasis - in most cases I think this is unnecessary since the text is largely clear and the novelty of the findings is obvious and need not be overstated.

Thank you for the comment. We have removed all extraneous italicization as suggested.

P4-5 lines 95-98 - It is unclear to me why the authors describe the second part of their screen as being in the opposite direction to the first (p4, lines 95-98), since the readout is the same (suppression by a query effector of growth restriction caused by another effector). This should be clarified in the text.

To clarify, our original intent was to indicate that effector-effector suppression can be identified in one of two ways: either a query is toxic and one or more strains on the array rescues that toxicity, or an array strain inhibits growth but is suppressed by co-expression of a non-toxic query. We have rephrased the original statement to precisely describe what we were looking for and to avoid the confusion of characterizing this as being in the “opposite direction”:

“Due to technical reasons, growth suppression was not always observed in both directions (when the identity of an array and query strain were reversed). While infrequent, such instances are likely due to the potential of epitope tags and spontaneous mutations within the yeast genome to mask some interactions. As such, we also looked for additional suppressors in which expression of a query gene was able to suppress growth inhibition caused by one of the IDTS on the array

(Figure 1C, Appendix Figure S2).” P5 line 103 suggests that several effector pairs are genetically unlinked (presumably based upon their physical distance on the chromosome) - however it would be useful to examine if there are obvious shared regulatory/promoter motifs in these pairs, or if they are otherwise co-regulated. For example are the pairs that appear to be largely similar in transcriptional dynamics (Figure S9C) also those closely placed on the genome (and thus possibly part of the same operon)? A cursory examination of figure 8A shows this to be the apparent case, except for mavQ and sidP. This merits further analysis and discussion.

Excluding the multiple SidE/SidJ paralogs in our dataset, of the effector-effector pairs we describe in this work, 6 are immediately adjacent to one another on the chromosome, 2 are nearby (within 2-3 loci), and 6 are completely unlinked. A statement to this effect is now included in the figure legend to Figure 8A. Notably, one of the strengths of our systematic approach to screening for these effector-effector relationships is that it unveiled several unlinked pairs as the published pairs (SidM/SidD; AnkX/Lem3; SidH/LubX) are all immediately adjacent to one another on the chromosome, which in some instances facilitated the discovery of the functional relationship between them.

To further examine whether regulatory coordination may be driving the linkages of effectors and their modulators, we used RockHopper (McClure R et al, 2013. *Nucleic acids research* **41**: e140; Tjaden B, 2015. *Genome biology* **16**: 1) to re-analyze the five RNA-seq Illumina datasets (>50 million reads from Weissenmayer BA et al, 2011 *PloS one* **6**: e17570) for evidence of operons.

These analyses indicate that of all the effector/modulator pairs that we have identified, only one pair (Ceg6-Lpg0208 and RavG-Lpg0210) are predicted to fall within the same operon, arguing against linkage as a driving mechanism to coordinate the co-expression of most of the functional pairs.

What else might be influencing linkage between an effector and its metaeffector/antagonist on the chromosome? One contributing factor may be that linkage ensures the concomitant segregation of both the effector and the modulator during instances of large-scale genomic rearrangements. Indeed, the possibility for genomically distant effectors to be split up is real as frequent large-scale genetic exchange has been observed in *Legionella pneumophila* (McAdam PR et al, 2014. *Genome Biol* **15**: 504; Sanchez-Buso L et al, 2014. *Nat Genet* **46**: 1205-1211).

One interesting focus of future study will be to determine whether the evolutionary distinctions between direct and indirect functional antagonists we report have implications for the types of genetic exchange that are likely to be tolerated by this pathogen.

With respect to regulatory motifs, several earlier studies have looked for patterns of effector regulation in *Legionella pneumophila* (Al-Khodori, S et al. *Infection and immunity* **77**, 374, (2009); Altman, E et al. *Journal of bacteriology* **190**, 1985, (2008); Feldheim, YS et al. *Molecular microbiology* **99**, 1059, (2016); Kessler, A et al. *Environmental microbiology* **15**, 646, (2013); Nevo, O et al. *Journal of bacteriology* **196**, 681, (2014); Rasis, M et al. *Molecular microbiology* **72**, 995, (2009); Sahr, T et al. *Molecular microbiology* **72**, 741, (2009); Tanner, JR et al. *Molecular microbiology* **100**, 1017, (2016); Tiaden, A et al. *Environmental microbiology* **12**, 1243, (2010); Zusman, T et al. *Molecular microbiology* **63**, 1508, (2007)). We examined each of these papers for obvious patterns of co-regulation between each effector and its direct or indirect antagonist. While a handful of effectors appear to be regulated by one or more signaling pathways that their cognate effectors are not, to fully explore this possibility and its implications to infection is likely a multi-

publication, multi-group effort outside the scope of this study.

Notably, however, of the effector/metaeffector pairs that we have examined at depth within the current manuscript, MavQ shows evidence of being regulated by the sensor kinase LqsT (Kessler, 2013) whereas its cognate metaeffector, SidP does not.

The RNAseq data in Figure S9 is mentioned in the discussion, I was unsure why this was not included more prominently in the results section. Also it is unclear as to what extent this supports the authors' suggestions that it shows "holding in check" of effector function. The data is all relative expression (on a per-gene basis), and so it is not apparent that even large relative changes represent absolute changes that might merit significant changes in post-transcriptional regulation. Of course my objection also presumes that RNA levels correlate with active, secreted effector levels outside of the bacteria, but this point also has some bearing upon the authors' interpretation.

The RNA-seq data was placed in the discussion largely because it is a focused re-analysis of others' published data (as indicated in the figure, from Weissenmayer, B.A., Prendergast, J.G., Lohan, A.J., and Loftus, B.J. (2011). PLoS One 6, e17570), highlighted as a next step towards placing our functional data into the context of the pathogen's intracellular life-cycle. Due to a shared appreciation for the limitations of these datasets that you raise, a more prominent place in the results would likely put a stronger emphasis on these preliminary observations than is warranted. Indeed, the relationship between the transcript levels of metaeffectors and their cognate effectors are not nearly as simple as what was observed by Nagai and colleagues (Kubori, 2009) for the first published pair (SidH/LubX). The impact on changes in relative transcript levels (versus absolute transcript levels) really depends on the mechanism of antagonism (steric hindrance or catalytic inactivation) and the relationship between bacterial transcription and protein translocation efficiency. Within this context, these data represent an opening discussion about what comes next – detailed proteomic analysis of effector protein levels and localization within the host cell. Such studies will be backed by the systems-level data we provide, but will require a considerable amount of optimization and methods/reagent development (such as custom antibodies for effectors of interest) outside the scope of the current study. We have included the following text to the discussion to address these points:

“To fully explore the regulatory network of effectors and metaeffectors during infection, detailed proteomic analysis of effector protein levels and localization within the host cell are obvious next steps for the field. Notably, the mechanisms of inhibition we describe are likely to be critically important for interpreting these results, as metaeffectors such as LegL1 that rely upon steric hindrance of their cognate effectors will require absolute protein levels greater than catalytic antagonists such as LupA.”

P6 line 130/131 - It would be useful to describe a little of what is known about MavQ (or even merely to state it is an effector of unknown function).

Indeed, MavQ is an effector of unknown function, which is now stated clearly in the text.

P7 line 148/149 - Some of the text is underlined, reasons unclear.

Removed as suggested above. Thank you.

Figure 5C - The authors describe the activity of the H183A mutant as being below the detection limit of the assay - they should show the detection limit on the graphs in 5B and C. Alternatively, if they mean that the results were not statistically different to no enzyme controls, then the appropriate statistical test should be described and p values (or other measure) shown on the graphs.

The key finding from Figure 5B are that ubiquitin is the preferred substrate of LupA. The figure legend has been modified to clarify this point: “The activity of LupA against Ubiquitin-AMC is significantly different than its activity against each of the other Ubiquitin-like substrates as assessed by unpaired, two-tailed Student's T tests (Ub vs. Nedd8: P value = 0.005; SUMO-1: P value = 0.005; SUMO-2: P value = 0.005; SUMO-3: P value = 0.005; n=2).”

The key finding of Figure 5C is that the H183A mutation ablates LupA activity against ubiquitin relative to the wild-type control. The figure legend has been modified to clarify this point: “Mutation of a predicted catalytic residue (H183) almost completely abolishes the *in vitro* hydrolase

activity. In the fluorescence-based assay described above, the mutant activity is significantly reduced from the wild-type enzyme activity as assessed by an unpaired, two-tailed Student's T test (P value = 0.000009, n=3)."

P10 line 222 - Cross-species rescue was not maintained for both gene pairs; this is correctly stated in line 225.

Thank you for catching this mistake. The phrase now reads, "As predicted, intra-species rescue was maintained for both gene pairs (Figure 6E, F)."

P10 line 241 - "Cross-species rescue" should be changed to "cross-species synergy", since no suppression/rescue occurs.

Thank you for catching this important distinction. We have made the correction as suggested.

P10-11 lines 234-254 - This section of the paper feels quite preliminary and in contrast to much of the detailed analysis elsewhere it seems the authors are stretching towards a mechanism, but failing to demonstrate one. Whilst I will agree if the authors state that this mechanism is beyond the scope of the paper, I feel that this section feels like a slight anticlimax before moving into the discussion. However, the discovery of such a stark synergy is very interesting and so I do feel a slight re-emphasis of this section would help increase the readability of the paper.

Thank you for the comments and for nevertheless appreciating the importance of highlighting these preliminary data. We agree completely. The section has been re-emphasized as suggested, with the goal of highlighting the preliminary nature of the studies relative to the rest of the paper and ending on a more robust statement of importance.

P 18 Line 415 - "(C) A predicted catalytic residue (H183) abolished the ... activity" should read "(C) Mutation of a predicted ..."

Thank you for the suggestion. Changed as noted.

The authors might consider swapping Figures 7B and C, since C is referred to in the text prior to B.

Thank you for the suggestion. Changed as noted.

In Figure S9 lubX is referred to as legU2 - this should be changed to be consistent with the text. Similarly for legc7/yf1A in figure 8A.

Thank you for the suggestion. Fig S9 has been changed as noted. Figure 8A was kept the same (ylfA) - to be consistent with Figure 2, but Figure 1B was changed to read "ylfA/legC7." These changes now reflect the consistent nomenclature throughout the rest of the text.

Thank you again for sending us your revised manuscript. We are now satisfied with the modifications made and I am pleased to inform you that your paper has been accepted for publication.

Corresponding Author Name: Alexander W. Ensminger

Manuscript Number: MSB-16-7381